# Effect of nonsense-mediated mRNA decay factor SMG9 deficiency on premature aging in zebrafish

Shaohong Lai[1,7], Hiroshi Shiraishi[1,7], Wulan Apridita Sebastian[2], Nobuyuki Shimizu [1], Ryohei Umeda[3], Mayo Ikeuchi[1], Kyoko Kiyota[1], Takashi Takeno[1], Shuya Miyazaki[1], Shinji Yano[4], Tatsuo Shimada[5], Akihiko Yoshimura[6], Reiko Hanada[3] & Toshikatsu Hanada [1] ✉

SMG9 is an essential component of the nonsense-mediated mRNA decay (NMD) machinery, a quality control mechanism that selectively degrades aberrant transcripts. Mutations in SMG9 are associated with heart and brain malformation syndrome (HBMS). However, the molecular mechanism underlying HBMS remains unclear. We generated *smg9* mutant zebrafish (*smg9*[oi7/oi7]) that have a lifespan of approximately 6 months or longer, allowing for analysis of the in vivo function of Smg9 in adults in more detail. *smg9*[oi7/oi7] zebrafish display congenital brain abnormalities and reduced cardiac contraction. Additionally, *smg9*[oi7/oi7] zebrafish exhibit a premature aging phenotype. Analysis of NMD target mRNAs shows a trend toward increased mRNA levels in *smg9*[oi7/oi7] zebrafish. Spermidine oxidase (Smox) is increased in *smg9*[oi7/oi7] zebrafish, resulting in the accumulation of byproducts, reactive oxygen species, and acrolein. The accumulation of *smox* mRNA due to NMD dysregulation caused by Smg9 deficiency leads to increased oxidative stress, resulting in premature aging.

Nonsense-mediated mRNA decay (NMD) is an evolutionarily conserved mRNA quality control mechanism that selectively degrades aberrant transcripts derived from mutant genes harboring premature translation termination codons (PTCs)[1]. In addition, at least 10% of normal mRNAs containing long 3′-untranslated region (3′-UTRs), introns in the 3′-UTR, and upstream open reading frames (uORFs) are degraded by NMD in response to environmental changes[2–4]. NMDs are involved in development, cell cycle regulation, DNA damage response, and immune system function. Disruption of NMD leads to nonsense or frameshift mutations that introduce PTCs, resulting in hereditary diseases, neurological disorders, immune disorders, and certain types of cancers[5,6].

SMG9 is an essential component of the NMD pathway. The SURF surveillance complex, composed of SMG1, UPF1, eRF1, and eRF3 in human cells, recognizes PTC on spliced mRNAs[7]. SMG1 phosphorylates UPF1 when an exon junction complex (EJC) is present at least 50-55 nucleotides downstream of the PTC[8]. SMG1-mediated UPF1 phosphorylation is a prerequisite and the commitment step in NMD[9]. SMG8 and SMG9 are associated with SMG1 to suppress the kinase activity of SMG1 until the SMG1-UPF1 complex joins the EJC[5]. Although SMG8 plays a role in suppressing

SMG1 activity, SMG9 presumably mediates the interaction between SMG8 and SMG1[10]. Given that SMG9 knockdown using siRNA resulted in the accumulation of phosphorylated UPF1, SMG9 is presumably essential for suppressing phosphorylated UPF1 and subsequent NMD pathway[5]. Moreover, NMD itself is positively regulated by SMG8/9 in vivo[5]. Therefore, how the dysfunction of SMG9 affects the NMD pathway remains unclear[5,10].

Autosomal recessive mutations in SMG9 are associated with heart and brain malformation syndrome (HBMS)[2]. To date, 12 patients from eight unrelated families carrying homozygous pathogenic SMG9 mutations have been reported[2,11–13]. However, little is known about how pathogenic SMG9 mutations affect the NMD pathway and why they cause HBMS. Moreover, global transcriptome analysis of SMG9-deficient patient cells showed no evidence of NMD impairment despite an abnormal transcriptional profile compared to that in control individuals[2].

Creating an in vivo model of SMG9 genetic modifications would help clarify the relationship between SMG9 mutations and pathogenesis in vivo. SMG9 knockout mice show abnormalities in brain, heart, and eye development[2]. However, SMG9 knockout mice displayed early embryonic lethality, making it difficult to analyze the pathogenesis of SMG9 in vivo.

[1]Department of Cell Biology, Oita University Faculty of Medicine, Yufu, Oita, Japan. [2]Department of Pediatrics, Oita University Faculty of Medicine, Yufu, Oita, Japan. [3]Department of Neurophysiology, Oita University Faculty of Medicine, Yufu, Oita, Japan. [4]Institute for Research Management, Oita University, Yufu, Oita, Japan. [5]Oita Medical Technology School, Japan College of Judo-Therapy, Acupuncture & Moxibustion Therapy, Oita, Japan. [6]Department of Microbiology and Immunology, Keio University School of Medicine, Tokyo, Japan. [7]These authors contributed equally: Shaohong Lai, Hiroshi Shiraishi. ✉e-mail: thanada@oita-u.ac.jp

Here, we report the generation and phenotypic analysis of *smg9* null mutant zebrafish (*smg9*[oi7/oi7]). These zebrafish showed brain and heart defects, as observed in SMG9-mutant patients. We also showed that the dysregulation of NMD due to Smg9 deficiency resulted in increased spermine oxidase (*smox*) mRNA, a PTC-containing mRNA. Overexpression of Smox induces reactive oxygen species (ROS) production, resulting in a premature aging phenotype in *smg9*[oi7/oi7] zebrafish.

## Results

### Generation of *smg9 mutant* zebrafish

To examine the tissue distribution of Smg9 in zebrafish, we analyzed *smg9* mRNA expression in various zebrafish tissues at 3 months of age and compared it with that in human tissues (Supplementary Fig. 1). *smg9* mRNA is broadly expressed in various tissues with a pattern similar to that in humans, except for the testes. To clarify the pathophysiological relationship between HBMS and SMG9, we generated *smg9* mutant zebrafish using CRISPR/Cas9 genome editing[14,15]. We targeted a gRNA located in the first exon to create a deletion in the *smg9* allele, resulting in a frameshift mutation with a 10-base pair deletion that likely corresponded to a null allele (Fig. 1a). Zebrafish carrying a frameshift mutation in the targeted exon were successfully obtained. We registered this *smg9* mutant zebrafish with ZFIN as *smg9*[oi7/oi7] and refer to it as such. Western blot analysis confirmed the absence of Smg9 protein in *smg9*[oi7/oi7] (Fig. 1b). A minor isoform reported in Ensembl was not observed, indicating that there was no expression of this protein isoform in zebrafish (Supplementary Fig. 2a).

Each zebrafish was genotyped by polymerase chain reaction (PCR) using genomic DNA (Supplementary Fig. 2b). *smg9*[oi7/oi7] embryos hatched at normal Mendelian ratios (Fig. 1c). The *smg9*[oi7/oi7] zebrafish exhibited a normal overall morphology until 5 months of age. No significant differences in body length were observed between *smg9*[oi7/oi7] and wild-type siblings at either the larval or adult stages (Fig. 1d–g).

### Brain malformation in *smg9*[oi7/oi7] larvae

We first evaluated the brains of *smg9*[oi7/oi7] larvae to examine the pathological similarities between HBMS and *smg9*[oi7/oi7] zebrafish. The telencephalon, the most anterior part of the brain, was significantly smaller in *smg9*[oi7/oi7] larvae than in wild-type siblings (Fig. 1h, i). Next, we examined proliferating radial glial progenitor cells that express S100β and proliferating cell nuclear antigen (PCNA), which act as neural stem cells in the cerebral cortex[16]. *smg9*[oi7/oi7] zebrafish showed significantly fewer radial glial progenitor cells in the dorsal telencephalon (Fig. 1j, k). Importantly, the reduced brain size of *smg9*[oi7/oi7] zebrafish was restored by exogenous Smg9 supplementation via *smg9* mRNA injection into *smg9*[oi7/oi7] embryos (Fig. 1l), indicating that the defects in brain development in *smg9*[oi7/oi7] zebrafish were caused by *smg9* deletion.

Furthermore, we examined malformations in the adult brain at 3 months post fertilization (mpf) and 6 mpf. The telencephalon was more affected than the other regions of the *smg9*[oi7/oi7] zebrafish brain at 3 mpf (Fig. 1m, n). At 6 mpf, all regions were significantly smaller in *smg9*[oi7/oi7] zebrafish than in their wild-type siblings, indicating that *smg9*[oi7/oi7] brains were affected (Fig. 1o, p).

Next, we assessed the role of Smg8 in *smg9*[oi7/oi7] zebrafish. SMG8 and SMG9 form a complex with SMG1 and suppress the kinase activity of SMG1 in human cells[17]. Homozygous mutations in *SMG8* cause Alzahrani–Kuwahara syndrome and heart, eye, and brain malformation syndromes, indicating an important role of the NMD machinery in humans[18]. Notably, *smg8* mutant (*smg8*[oi8/oi8]) zebrafish, in which wild-type smg8 mRNA is absent, did not exhibit any phenotype (Supplementary Fig. 3a–c and Supplementary Fig. 4a, b). Furthermore, the Smg8 mutation in *smg9*[oi7/oi7] zebrafish did not affect the phenotype (Supplementary Fig. 4a, b), indicating that Smg8 is not critical for the pathogenesis of *smg9*[oi7/oi7] zebrafish. Some NMD factors are regulated by an autoregulatory negative feedback mechanism[19]. However, *smg9* mRNA levels in *smg8*[oi8/oi8] zebrafish were not increased (Supplementary Fig. 5), indicating that the absence of a *smg8*[oi8/oi8] phenotype was not caused by a compensatory upregulation of Smg9.

### Impaired cardiac contraction in *smg9*[oi7/oi7] zebrafish

Given that heart malformations, such as interrupted aortic arch, hypoplastic valves, and ventricular septal defects, have been reported in humans with mutant *SMG9*[2], we next assessed the morphology and function of *smg9*[oi7/oi7] hearts. *smg9*[oi7/oi7] larvae showed no difference in heart size and overall morphology, including the thickness of the myocardium and development of the atrioventricular septum, compared to wild-type siblings (Fig. 2a–c). Cardiac contractions were significantly reduced in *smg9*[oi7/oi7] larvae (Fig. 2d, e, and Supplementary Movie), although there was no difference in heart rate between *smg9*[oi7/oi7] and wild-type larvae (Fig. 2f). The reduced cardiac contraction in *smg9*[oi7/oi7] zebrafish was restored by exogenous Smg9 supplementation via *smg9* mRNA injection into *smg9*[oi7/oi7] embryos (Fig. 2g), suggesting that cardiac dysfunction in *smg9*[oi7/oi7] zebrafish was caused by *smg9* deletion. Thus, Smg9 deficiency results in impaired cardiac function but does not affect morphogenesis in the developing hearts of zebrafish.

### Premature aging in *smg9*[oi7/oi7] zebrafish

Although Smg9 knockout mice exhibit embryonic lethality, *smg9*[oi7/oi7] zebrafish grow into adulthood, facilitating further investigation of Smg9 deficiency in vivo[2]. *smg9*[oi7/oi7] zebrafish had a reduced lifespan, with approximately half of the *smg9*[oi7/oi7] zebrafish dying before 6 months of age (Fig. 3a). The heterozygous *smg9*[oi7/+] zebrafish did not exhibit any notable phenotypes or differences in lifespan compared with their wild-type siblings (Fig. 3a).

Notably, *smg9*[oi7/oi7] zebrafish displayed aging phenotypes at 6 months of age, such as ruffled appearance and kyphosis observed during normal aging in zebrafish[20], although these were not obvious at 3 months of age (Fig. 3b, c).

Histologically, thinner intestinal epithelium was observed in *smg9*[oi7/oi7] zebrafish at 6 months of age (Fig. 3d). We observed reduced proliferating PCNA-positive intestinal stem cells in *smg9*[oi7/oi7] zebrafish (Fig. 3e). Furthermore, γH2AX, a marker of DNA damage, was increased in the intestinal epithelium of *smg9*[oi7/oi7] zebrafish at 6 months of age (Fig. 3f). Cleaved caspase-3, a marker of apoptotic cells, was also increased in the epithelium of *smg9*[oi7/oi7] zebrafish (Fig. 3g). A decline in nephron number and tubular atrophy were observed in the kidneys of *smg9*[oi7/oi7] zebrafish (Fig. 3h). Thinner skin epithelial layers and muscle fiber bundles were also observed in *smg9*[oi7/oi7] zebrafish (Fig. 3i, j). These data indicated that *smg9*[oi7/oi7] zebrafish display an aging phenotype.

Infertility is an age-related phenotype[21]. Decreased fertility has been observed in *smg9*[oi7/oi7] zebrafish. Infertility in *smg9*[oi7/oi7] zebrafish worsened with age (Fig. 3k). The size of the *smg9*[oi7/oi7] zebrafish testes was comparable to that of the wild-type until 3 months of age. However, the *smg9*[oi7/oi7] zebrafish testes were smaller and thinner than those of wild-type zebrafish at 6 months of age (Fig. 3l). Histological examination of the testes revealed a decreased number of mature sperm cells in *smg9*[oi7/oi7] zebrafish (Fig. 3m). These data suggest that Smg9 deficiency leads to premature aging.

To confirm the premature aging phenotype, we assessed senescence-associated β-galactosidase activity (SA-β-gal), a histochemically detectable biomarker of senescence in organismic aging[22]. At one month and three months of age, no difference in SA-β-gal activity in the skin was found between *smg9*[oi7/oi7] zebrafish and their wild-type siblings (Fig. 4a–d). However, the activity increased in *smg9*[oi7/oi7] zebrafish at 6 months of age (Fig. 4e, f). We examined SA-β-gal activity directly in sliced sections of the dissected brains to study neural aging. We did not find a significant difference in SA-β-gal activity in the brain at 1 month (Fig. 4g, h; Supplementary Fig. 6a). However, a significant increase in SA-β-gal activity was observed at 3 months (Fig. 4i, j; Supplementary Fig. 6b) and 6 months (Fig. 4k, l; Supplementary Fig. 6c) of age. We also observed a marked reduction in HuC/D-stained mature neurons in the telencephalons of *smg9*[oi7/oi7] zebrafish at 6 months of age (Supplementary Fig. 7a, b). To investigate the physiological output of mature neuron loss, we analyzed the locomotor activity and acoustic stimulation responses of *smg9*[oi7/oi7] zebrafish. The *smg9*[oi7/oi7] zebrafish showed a significant decrease in locomotor

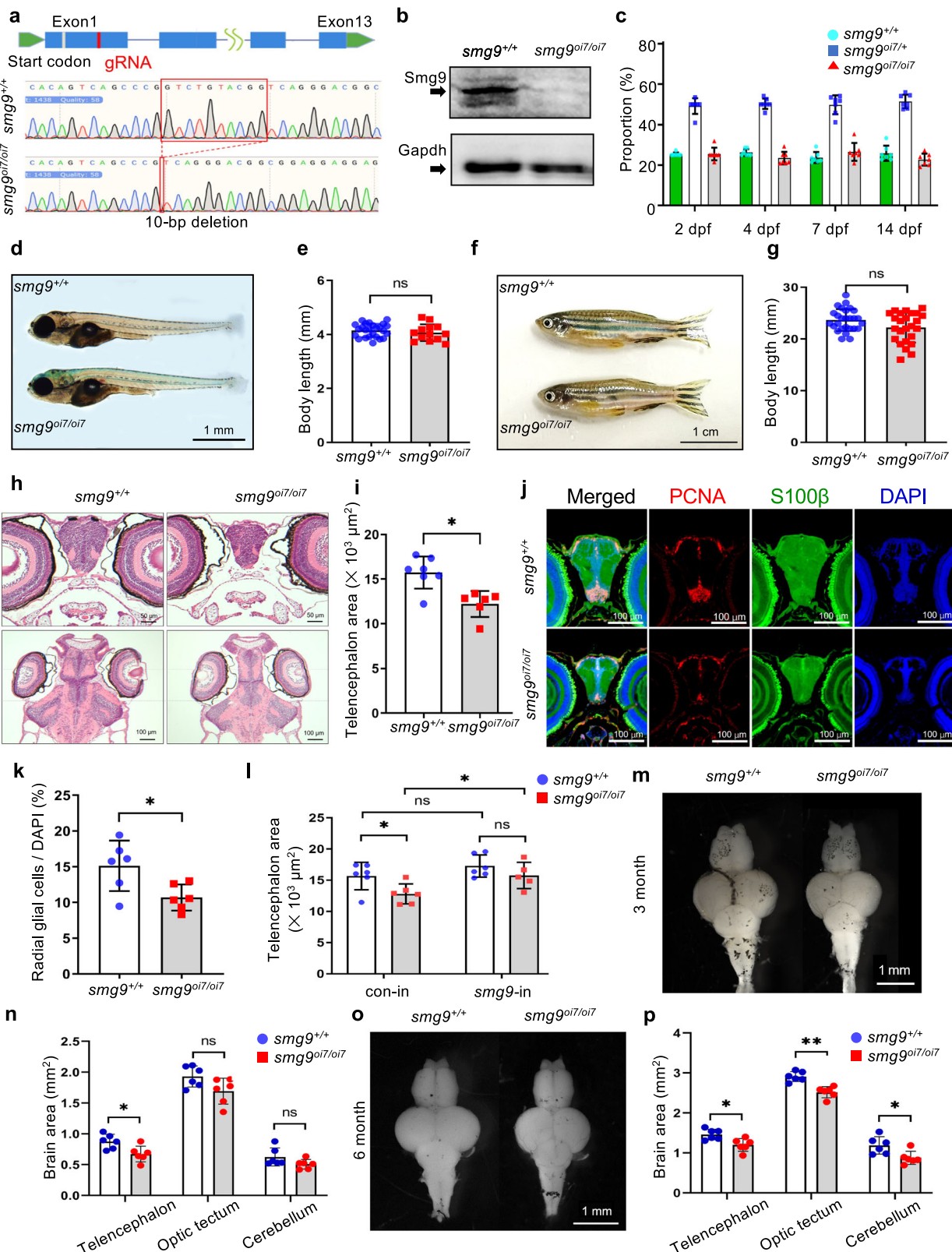

activity and acoustic stimulation response compared to their wild-type siblings at 3 months of age (Supplementary Fig. 7c, d). The difference was even greater at 6 months, with or without stimulation (Supplementary Fig. 7e, f). To confirm senescence in *smg9*[oi7/oi7] zebrafish, the mRNA expression levels of several senescence markers, *p21*, *p53*, *p16*, *p27*, and *p57*, were evaluated using reverse-transcription quantitative polymerase chain reaction (RT-qPCR). Among these, *p21*, *p53*, and *p57* mRNA levels were significantly higher in *smg9*[oi7/oi7] zebrafish than in their wild-type siblings (Fig. 4m). The senescence-associated secretory phenotype (SASP) was also assessed; significant increases in *Il-6* and *Il-1β* were observed in *smg9*[oi7/oi7] zebrafish (Fig. 4n). Thus, Smg9 deficiency resulted in premature aging in zebrafish.

**Fig. 1 | Smg9-deficient zebrafish display brain malformations. a** Genomic locus diagram depicting the zebrafish *smg9* gene and its 10-nucleotide deletion using CRISPR/Cas9. The red box indicates the mutated region in *smg9*[oi7/oi7] zebrafish. **b** Western blot analyses of lysates from *smg9*[+/+] and *smg9*[oi7/oi7] larvae at 2 dpf to determine the Smg9 protein levels. **c** The percentage of *smg9*[+/+], *smg9*[oi7/+], and *smg9*[oi7/oi7] zebrafish were determined from 2 to 14 dpf. **d** Representative images of *smg9*[+/+] and *smg9*[oi7/oi7] larvae at 14 dpf. Scale bar: 1 mm. **e** Body length of *smg9*[+/+] and *smg9*[oi7/oi7] larvae at 14 dpf. **f** Representative images of *smg9*[+/+] and *smg9*[oi7/oi7] fish at 3 mpf. Scale bar: 1 cm. **g** Body length of *smg9*[+/+] and *smg9*[oi7/oi7] zebrafish at 3 mpf. **h** Representative images of H&E staining of transverse and coronal sections of *smg9*[+/+] and *smg9*[oi7/oi7] larvae at 14 dpf. Scale bar: 50 µm in the upper panel and 100 µm in the lower panel. **i** Telencephalon areas of *smg9*[+/+] and *smg9*[oi7/oi7] larvae at 14 dpf were measured using transverse sections of H&E images. **j** Representative images of immunostaining of 14 dpf larval brain tissue with PCNA (red), S100β (green), and DAPI (blue). Scale bar: 100 µm. **k** Graph shows PCNA/S100β double-positive radial glial cells among DAPI-positive cells in the telencephalon of *smg9*[+/+] and *smg9*[oi7/oi7] larvae at 14 dpf. **l** Telencephalon size of 14 dpf *smg9*[+/+] and *smg9*[oi7/oi7] larvae injected with control (con-in) or zebrafish *smg9* mRNA (*smg9*-in). **m** Representative images of the dissected brains of *smg9*[+/+] and *smg9*[oi7/oi7] zebrafish at 3 mpf. Scale bar: 1 mm. **n** Quantification of the areas of the telencephalon, optic tectum, and cerebellum in *smg9*[+/+] and *smg9*[oi7/oi7] zebrafish at 3 mpf. **o** Representative images of dissected brains of *smg9*[+/+] and *smg9*[oi7/oi7] zebrafish at 6 mpf. Scale bar: 1 mm. **p** Quantification of the areas of the telencephalon, optic tectum, and cerebellum in *smg9*[+/+] and *smg9*[oi7/oi7] zebrafish at 6 mpf. Error bars indicate SD. *$P < 0.05$, **$P < 0.01$, and ***$P < 0.001$. ns: not significant.

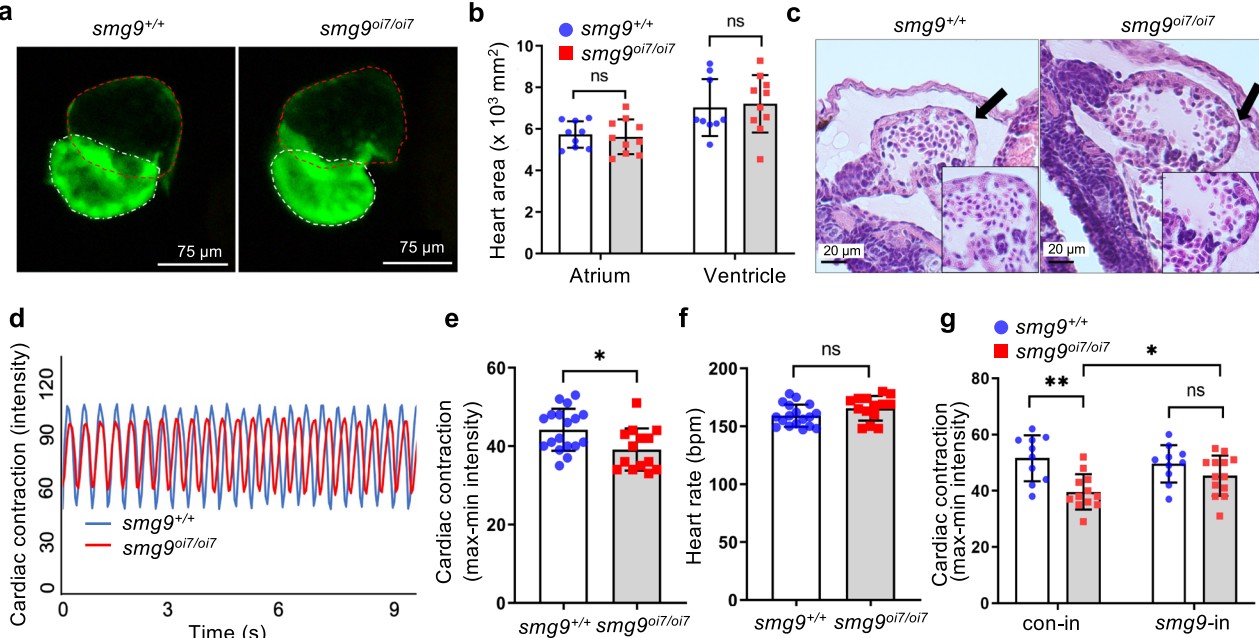

**Fig. 2 | Impaired cardiac contractions in Smg9-deficient zebrafish.**
**a** Representative image of the heart of *cmlc2:EGFP* zebrafish. The red and white dotted lines indicate the atrium and ventricle, respectively. **b** Quantification of the atrium and ventricle areas in *smg9*[+/+] and *smg9*[oi7/oi7] zebrafish at 6 dpf. **c** Representative images of hematoxylin and eosin staining of *smg9*[+/+] and *smg9*[oi7/oi7] hearts at 6 dpf. **d** Representative dynamic pixel change patterns of the heart in *cmlc2:EGFP* zebrafish, showing the heartbeat and cardiac contraction of *smg9*[+/+] and *smg9*[oi7/oi7] zebrafish. **e** Quantification of cardiac contractions in *smg9*[+/+] and *smg9*[oi7/oi7] zebrafish larvae at 6 dpf. **f** Quantification of the heart rates of *smg9*[+/+] and *smg9*[oi7/oi7] zebrafish larvae at 6 dpf. **g** Quantification of cardiac contractions of 6 dpf *smg9*[+/+] and *smg9*[oi7/oi7] larvae injected with control (con-in) or zebrafish *smg9* mRNA (*smg9*-in). Error bars indicate SD. bpm: beats per minute. *$P < 0.05$, **$P < 0.01$, and ***$P < 0.001$. ns: not significant.

Although exogenous Smg9 supplementation via *smg9* mRNA injection into *smg9*[oi7/oi7] embryos restored the brain and heart phenotypes at the larval stage, it did not ameliorate premature aging phenotypes, such as lifespan, fertility, and SA-β-gal activity (Supplementary Fig. 8a–c). Because the premature aging phenotype in *smg9*[oi7/oi7] zebrafish is observed at the adult stage, the long-term expression of exogenous Smg9 is required to elucidate the effect of exogenous Smg9 expression on premature aging phenotypes.

**Smg9 deficiency impairs the NMD pathway in zebrafish**
To assess the effects of Smg9 deficiency on the NMD pathway, we compared the expression levels of several mRNAs affected by Upf1 deficiency in zebrafish. The transcript levels of these genes are likely regulated by the NMD machinery[23]. We used RT-qPCR to analyze the mRNA levels of serine/arginine-rich splicing factor 3a (*srsf3a*), serine/arginine-rich splicing factor 5a (*srsf5a*), ribosomal protein l22 like 1 (*rpl22l1*), and ribosomal protein L10a (*rpl10a*). Transcript levels were elevated in all three except for *srsf5a*, suggesting impairment of the NMD pathway in *smg9*[oi7/oi7] larvae (Fig. 5a–d). We examined six orthologs of mouse NMD target genes in *smg9*[oi7/oi7] zebrafish (Fig. 5e–j)[24]. The mRNA levels of growth arrest and

DNA damage-inducible protein 45β (*gadd45b2*), cysteinyl-tRNA synthetase (*cars*), p53, DNA damage-regulated 1 (*pdrg1*), and DNA damage-inducible transcript 3 (*ddit3*) were significantly increased in *smg9*[oi7/oi7] larvae (Fig. 5e, g–i). Although the mRNA levels of activating transcription factor 4 (*atf4*) and Ras association domain family member 1 (*rassf1*) were not significantly increased, they tended to increase compared to those in wild-type siblings (Fig. 5f, j). A tendency toward increased mRNA levels of these NMD-targeted candidate genes in *smg9*[oi7/oi7] larvae was also observed in adult zebrafish (Supplementary Fig. 9).

Phosphorylated UPF1 promotes the NMD pathway. To assess the effect of the SMG9 deficiency on UPF1 phosphorylation in vivo, we expressed the exogenous human UPF1 protein in *smg9*[oi7/oi7] larvae and examined its phosphorylation status. We used human UPF1 instead of zebrafish Upf1 because no antibodies recognized phosphorylated zebrafish Upf1. We confirmed that human UPF1 is a substrate for zebrafish Smg1 and that phosphorylation of human UPF1 was enhanced by zebrafish Smg1 in 293 T cells (Supplementary Fig. 10a).

We examined whether human and zebrafish SMG8 and SMG9 are interchangeable in the SMG1:SMG8:SMG9 complex. First, we

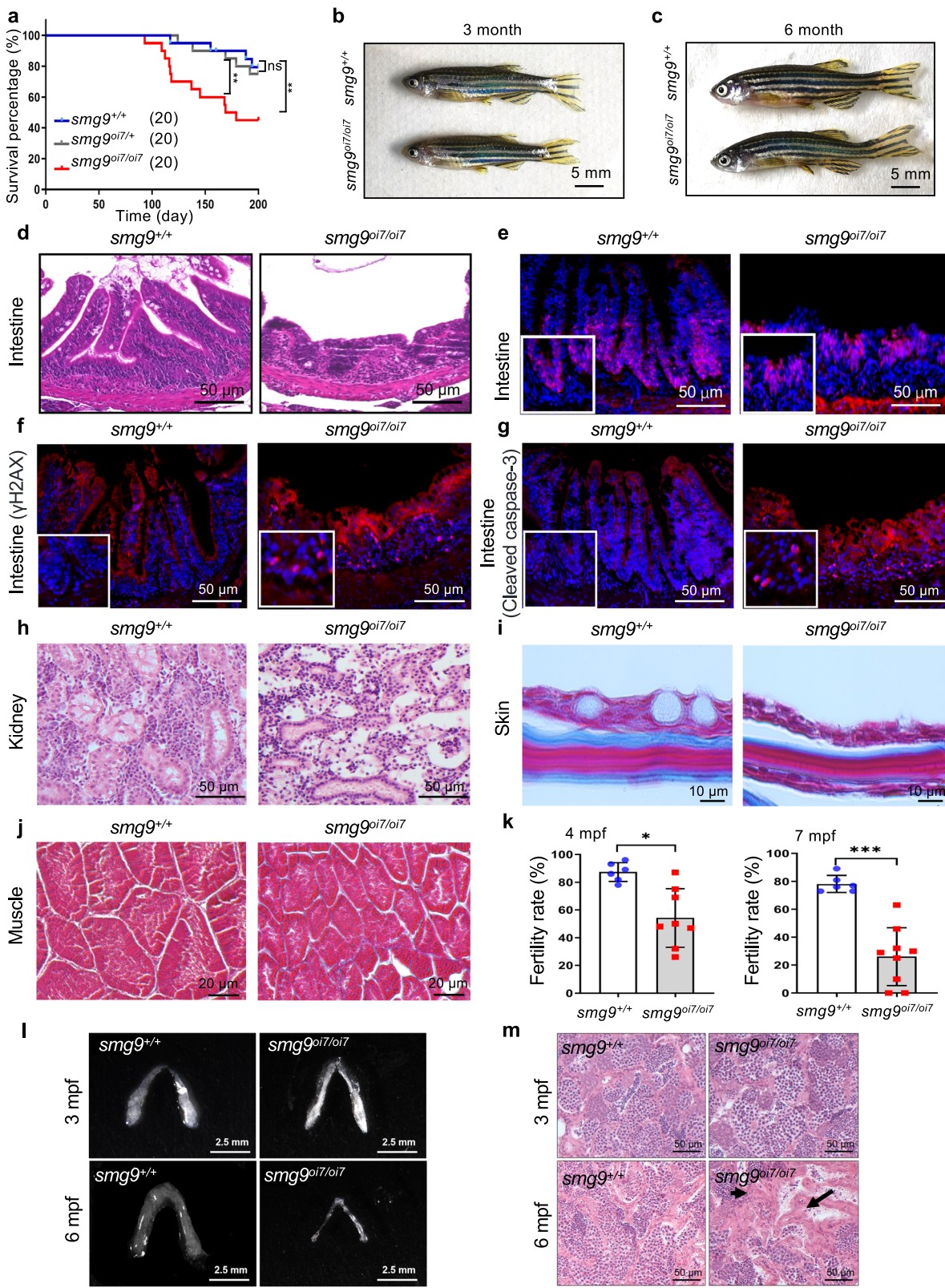

overexpressed the human SMG8:SMG9 heterodimers in 293 T cells. As reported[5], the human SMG8:SMG9 heterodimer suppressed SMG1 kinase activity in vitro. This effect was also observed upon overexpression of the zebrafish Smg8:Smg9 heterodimer. Furthermore, human SMG8 and zebrafish SMG9 heterodimers, and conversely, zebrafish SMG8 and human SMG9 heterodimers, inhibited the activity of human SMG1 in UPF1

phosphorylation in vitro (Supplementary Fig. 10b). In addition, *SMG9* mRNA injection into *smg9*[oi7/oi7] embryos restored the reduced brain size and cardiac contraction of *smg9*[oi7/oi7] zebrafish (Supplementary Fig. 11a–e). These results indicate that the components of the SMG1:SMG8:SMG9 complex are mutually interchangeable between humans and zebrafish. Indeed, SURF components are highly conserved in vertebrates[25], with more

**Fig. 3 | Premature aging phenotype in various tissues in Smg9-deficient zebrafish.** **a** Kaplan–Meier survival curve analysis showing the lifespan of $smg9^{+/+}$ ($n = 20$), $smg9^{oi7/+}$ ($n = 20$), and $smg9^{oi7/oi7}$ ($n = 20$) zebrafish. **b, c** Representative images of gross appearance of $smg9^{+/+}$ and $smg9^{oi7/oi7}$ zebrafish at 3 mpf and 6 mpf. Scale bar: 5 mm. **d** Representative H&E images of the intestine showing tissue morphology in $smg9^{+/+}$ and $smg9^{oi7/oi7}$ zebrafish at 6 mpf. Scale bar: 50 μm. **e–g** Representative images of immunofluorescence staining for PCNA (**e**), γH2AX (**f**), and cleaved caspase-3 (**g**) in the intestinal segments of $smg9^{+/+}$ and $smg9^{oi7/oi7}$ zebrafish at 6 mpf. Scale bar: 50 μm. **h** Representative H&E images of the kidney tissue morphology in $smg9^{+/+}$ and $smg9^{oi7/oi7}$ zebrafish at 6 mpf. Scale bar: 50 μm. **i, j** Representative images of Masson's trichrome staining of $smg9^{+/+}$ and $smg9^{oi7/oi7}$ skin (**i**) and muscle (**j**) at 6 mpf. Scale bar: 10 μm (**i**). Scale bar: 20 μm (**j**). **k** Fertility rate of $smg9^{+/+}$ and $smg9^{oi7/oi7}$ zebrafish at 4 mpf and 7 mpf, as determined by mating $smg9^{+/+}$ or $smg9^{oi7/oi7}$ male zebrafish with $smg9^{+/+}$ female zebrafish and counting embryo survival after 12 hpf. **l** Representative images of the morphological features of testes in $smg9^{+/+}$ and $smg9^{oi7/oi7}$ zebrafish at 3 mpf and 6 mpf. Scale bar: 2.5 mm. **m** Representative image of H&E staining of testes tissue in $smg9^{+/+}$ and $smg9^{oi7/oi7}$ at 3 mpf and 6 mpf. Sperm (black arrow) was absent in the 6 mpf $smg9^{oi7/oi7}$ zebrafish. Scale bar: 50 μm. Error bars indicate SD. *$P < 0.05$, ** $P < 0.01$, and *** $P < 0.001$. ns: not significant.

than 70% of the amino acids being conserved between zebrafish and humans. Structural analysis of the human SMG1 protein revealed that the domain of SMG1 important for regulating activity is conserved between zebrafish and humans.

In contrast to the suppressive function of the SMG8:SMG9 heterodimer in vitro, phosphorylation of UPF1 was weaker in $smg9^{oi7/oi7}$ larvae than in their wild-type siblings (Fig. 5k, l), suggesting that Smg9 promotes the phosphorylation of Upf1 in vivo.

### Accumulation of *smox* mRNAs in $smg9^{oi7/oi7}$ zebrafish

We examined glutathione peroxidase 3 (*gpx3*), spermine oxidase (*smox*), oxysterol-binding protein 2 (*osbp2*), and zinc finger protein 155 (*znf155*) transcripts, which are upregulated in patients with the *SMG9* homozygous missense mutation (Val184Ala mutation)[11]. *Smox* and *gpx3* mRNA levels were increased in $smg9^{oi7/oi7}$ zebrafish (Fig. 6a and Supplementary Fig. 12).

To examine the contribution of increased expression of *smox* and *gpx3* to $smg9^{oi7/oi7}$ zebrafish phenotypes, we microinjected *smox* or *gpx3* mRNAs into zebrafish single-cell-stage embryos and observed them at 14 dpf. Brain size decreased in the *smox* or *gpx3* mRNA-injected zebrafish (Fig. 6b, c and Supplementary Fig. 13). Cardiac contraction but not heart rate also decreased in *smox* mRNA-injected zebrafish (Fig. 6d, e), similar to that in $smg9^{oi7/oi7}$ zebrafish. *Smox* mRNA injection also increased SA-β-Gal levels in 4-month-old adult zebrafish brains (Fig. 6f), although it did not ameliorate other premature aging phenotypes, such as lifespan and fertility (Supplementary Fig. 14a, b). Thus, these results suggest that the increased levels of *smox* and *gpx3* mRNAs in $smg9^{oi7/oi7}$ contribute to both the brain and heart malformation phenotypes and the premature aging phenotype of $smg9^{oi7/oi7}$ zebrafish.

### ROS and acrolein accumulation in $smg9^{oi7/oi7}$ zebrafish

ROS and acrolein are the byproducts of the SMOX reaction. Smox catalyzes the conversion of spermine to spermidine. This reaction produces ROS and acrolein, which promote premature aging. First, we measured intracellular ROS production by staining cells dispersed from the brains of $smg9^{oi7/oi7}$ zebrafish and their siblings with a CellROX fluorogenic probe. As expected, ROS production was significantly increased in $smg9^{oi7/oi7}$ cells at both 3 and 6 months of age (Fig. 7a–d). Notably, ROS production increased with age, consistent with the severity of the disease phenotype.

Acrolein production was analyzed using an acroleinRED fluorescent dye probe. $smg9^{oi7/oi7}$ zebrafish showed high intensity of acrolein staining in $smg9^{oi7/oi7}$ zebrafish at 3 months of age (Fig. 7e, f). Their fluorescence intensity was further enhanced with age, as was ROS production at 6 months of age (Fig. 7g, h). These results indicated that increased Smox-induced ROS and acrolein production in $smg9^{oi7/oi7}$ zebrafish.

In addition to ROS, acrolein directly affects mitochondrial oxidative stress by increasing ROS production and decreasing glutathione content and aconitase activity, resulting in the impaired function of the mitochondrial electron transport system[26]. To examine mitochondrial damage, $smg9^{oi7/oi7}$ zebrafish brains at 6 months of age were examined using electron microscopy. The number of damaged mitochondria, characterized by swelling and vacuolization, was significantly increased in the brains of $smg9^{oi7/oi7}$ zebrafish (Fig. 7i, j). These results suggest that high Smox activity in $smg9^{oi7/oi7}$ zebrafish increased ROS and acrolein production, resulting in mitochondrial damage.

Smox catalyzes the conversion of spermine to spermidine, and spermidine prevents cell senescence, suppresses the occurrence or severity of age-related diseases, and prolongs lifespan (Supplementary Fig. 15a)[27,28]. At the same time, this reaction produces ROS and acrolein, which promote premature aging. The $smg9^{oi7/oi7}$ displayed high levels of ROS and acrolein. However, no protective effects were observed for spermine/spermidine (Supplementary Fig. 15b, c). These results suggested that byproducts play a dominant role in the premature aging phenotype.

### Rescue of brain and heart defects by SMOX and ROS inhibitors

To prove that Smox and its byproduct, ROS, are related to the pathogenesis of $smg9^{oi7/oi7}$ zebrafish, $smg9^{oi7/oi7}$ larvae were treated with either a SMOX or ROS inhibitor. MDL72527 is globally used as a SMOX inhibitor. MDL72527 treatment significantly restored the brain size (Fig. 8a, b). The proliferation of radial glial cells tended to be restored by MDL-72527 treatment in $smg9^{oi7/oi7}$ brains (Fig. 8c, d). Similarly, the ROS inhibitor N-acetyl-L-cysteine (NAC) significantly restored brain size (Fig. 8e, f). The number of proliferating radial glial cells in $smg9^{oi7/oi7}$ brains tended to be restored by NAC (Fig. 8g, h).

MDL-72527 significantly improved cardiac contractions in $smg9^{oi7/oi7}$ larvae (Fig. 8i, j), and NAC showed a tendency to improve cardiac contractions in $smg9^{oi7/oi7}$ larvae (Fig. 8k, l). However, the inhibitors did not mitigate premature aging phenotypes, including lifespan, fertility, or SA-β-gal activity (Supplementary Fig. 16a–c). The accumulation of smox and ROS produced after the juvenile stage might contribute to premature aging phenotypes, and treatment with inhibitors only during the juvenile stage is insufficient. The combination of Smox and ROS inhibitors was also effective in restoring the brain size (Supplementary Fig. 17). Simultaneously, to confirm the rescue effect of the Smox inhibitor, a Smox morpholino was used, which notably restored the size of the brain (Fig. 8m, n) and proliferation of radial glial cells (Fig. 8o, p) and improved cardiac contractions in the $smg9^{oi7/oi7}$ larvae (Fig. 8q, r).

In conclusion, these results demonstrate that Smox and its byproduct, ROS, play critical roles in mediating brain and heart defects in $smg9^{oi7/oi7}$ zebrafish.

## Discussion

In the present study, we aimed to elucidate the pathogenesis of HBMS in $smg9^{oi7/oi7}$ zebrafish. The $smg9^{oi7/oi7}$ zebrafish displayed congenital brain and heart abnormalities similar to human HBMS caused by an *SMG1* homozygous nonsense mutation. $smg9^{oi7/oi7}$ zebrafish show decreased numbers of radial glial progenitor cells and mature neurons and reduced brain size. In addition, cardiac contraction was significantly reduced in $smg9^{oi7/oi7}$ zebrafish compared with that in the wild-type. Additionally, $smg9^{oi7/oi7}$ zebrafish exhibit premature senescence, including a ruffled appearance and kyphosis, at 6 months of age. Analysis of mRNAs regulated by the NMD pathway showed a trend toward increased mRNA levels in $smg9^{oi7/oi7}$ zebrafish, indicating that Smg9 promotes the NMD pathway. Indeed, the phosphorylation of UPF1 was decreased in $smg9^{oi7/oi7}$ zebrafish. Notably, *smox* mRNA levels increased in $smg9^{oi7/oi7}$ zebrafish. In addition, ROS and acrolein, the toxic byproducts of SMOX-induced spermine catabolism, were increased in $smg9^{oi7/oi7}$ zebrafish. The SMOX and ROS inhibitors improved heart and brain abnormalities in $smg9^{oi7/oi7}$ zebrafish. The *smox* mRNA that accumulates due to NMD dysregulation caused by Smg9 deficiency leads to

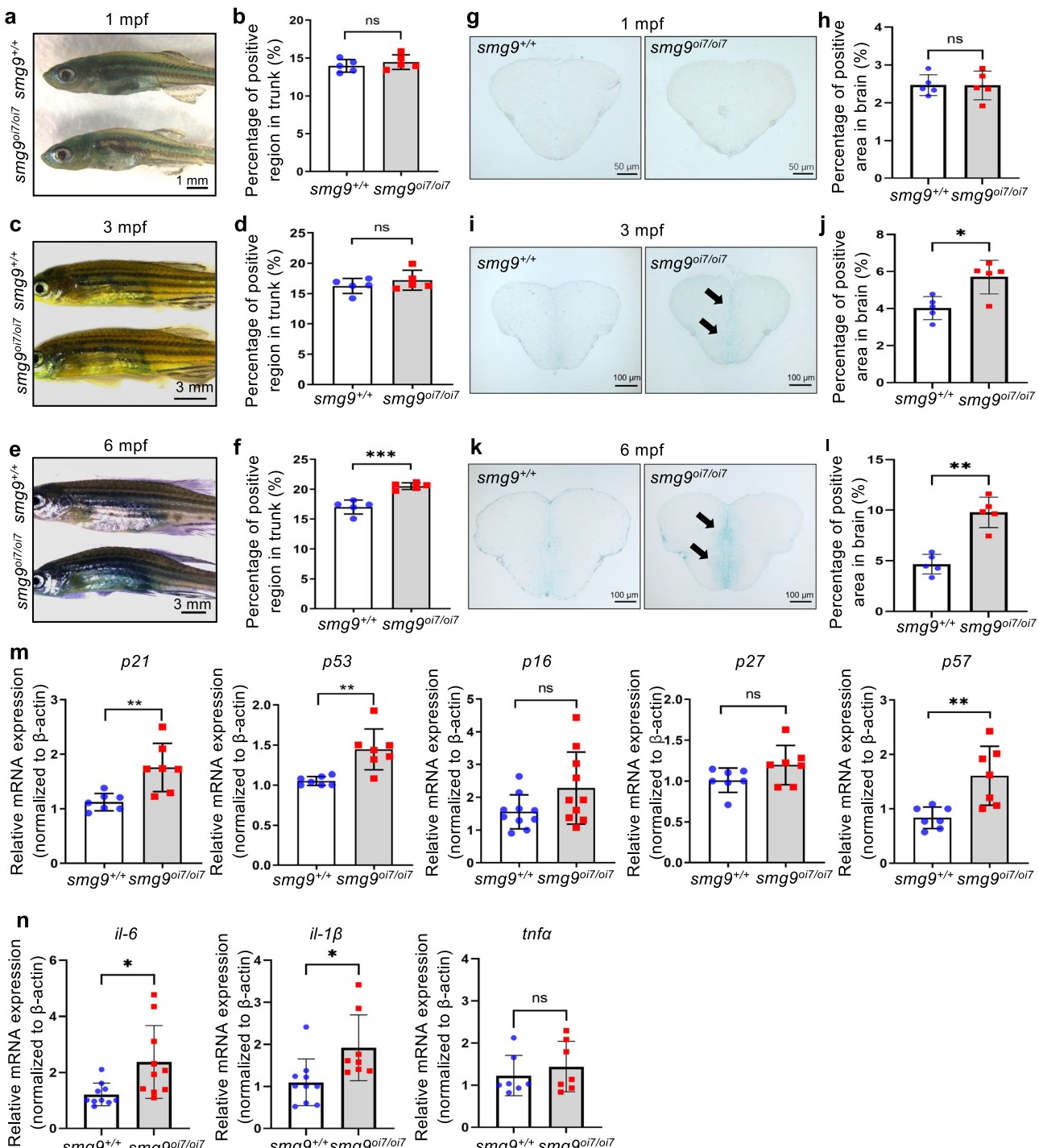

**Fig. 4 | Age-related staining and markers in Smg9-deficient zebrafish. a, b** SA-β-gal staining of the skin of *smg9*[+/+] and *smg9*[oi7/oi7] zebrafish at 1 mpf. Scale bar: 1 mm (**a**). Quantification of the percentage of positive areas in the trunk at 1 mpf (**b**). **c, d** SA-β-gal staining of the skin of *smg9*[+/+] and *smg9*[oi7/oi7] zebrafish at 3 mpf (**c**). Quantification of the percentage of positive areas in the trunk at 3 mpf. Scale bar: 3 mm (**d**). **e, f** SA-β-gal staining of the skin of *smg9*[+/+] and *smg9*[oi7/oi7] zebrafish at 6 mpf. Scale bar: 3 mm (**e**). Quantification of the percentage of positive areas in the trunk at 6 mpf (**f**). **g, h** SA-β-gal staining of the frozen brain sections of *smg9*[+/+] and *smg9*[oi7/oi7] zebrafish at 1 mpf. Scale bar: 50 μm (**g**). Quantification of the percentage of positive areas in the brain at 1 mpf (**h**). **i, j** SA-β-gal staining of the frozen brain sections of *smg9*[+/+] and *smg9*[oi7/oi7] zebrafish at 3 mpf. The arrows indicate positive regions. Scale bar: 100 μm (**i**). Quantification of the percentage of positive areas in the brain at 3 mpf (**j**). **k, l** SA-β-gal staining of the frozen brain sections of *smg9*[+/+] and *smg9*[oi7/oi7] zebrafish at 6 mpf. The arrows indicate positive regions. Scale bar: 100 μm (**k**). Quantification of the percentage of positive areas in the brain at 6 mpf (**l**). **m** Quantitative PCR analysis of *p21, p53, p16, p27,* and *p57* expression in *smg9*[+/+] and *smg9*[oi7/oi7] zebrafish larvae at 14 dpf. **n** Quantitative PCR analysis of *il-6, il-1β* and *tnfα* expression in *smg9*[+/+] and *smg9*[oi7/oi7] zebrafish larvae at 14 dpf. Error bars indicate SD. \*$P < 0.05$, \*\* $P < 0.01$, and \*\*\* $P < 0.001$. ns not significant.

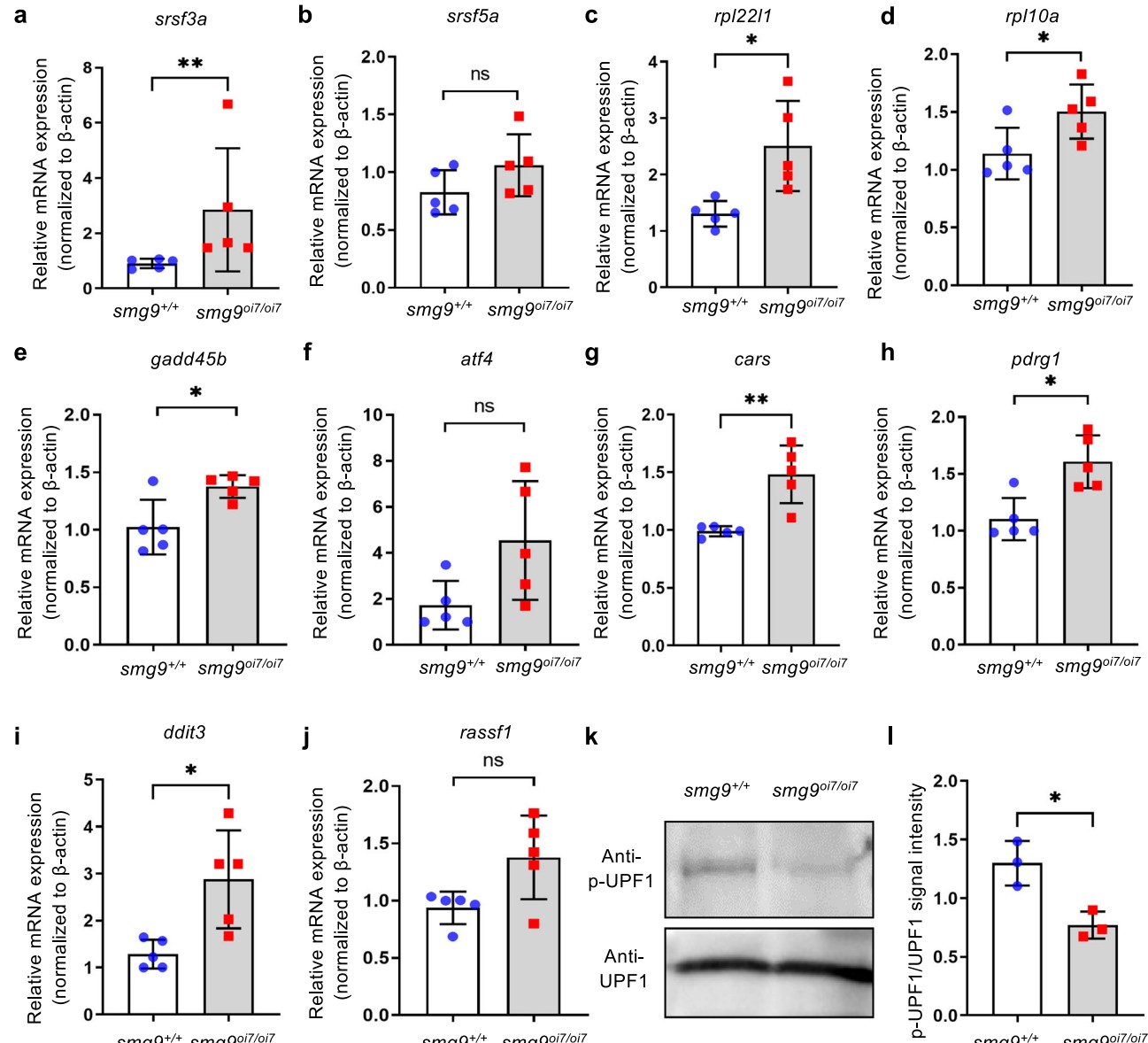

**Fig. 5 | Upregulation of endogenous targets of nonsense-mediated mRNA decay (NMD) in Smg9-deficient zebrafish. a–j** Quantitative PCR analysis of endogenous target genes of NMD in *smg9*[+/+] and *smg9*[oi7/oi7] larvae at 14 dpf. **k** Western blot analysis of lysates from *smg9*[+/+] and *smg9*[oi7/oi7] larvae at 2 dpf to determine the total FLAG-UPF1 (UPF1) and phosphorylated FLAG-UPF1(p-UPF1) levels. **l** Quantification of the total FLAG-UPF1 (UPF1) and phosphorylated FLAG-UPF1 (p-UPF1) levels in three independent experiments. Error bars indicate SD. \*$P < 0.05$, \*\* $P < 0.01$, and \*\*\* $P < 0.001$. ns: not significant.

increased production of ROS and acrolein, resulting in the pathogenesis of HBMS.

We successfully established Smg9 mutated zebrafish as a model of HBMS. In contrast, *Smg9*-null mutations in mice result in embryonic lethality due to placental defects, rendering detailed analyses difficult[2]. Although malformations of the brain, heart, and eyes have been observed, the phenotypes of *Smg9* knockout mice are variable and incompletely penetrant[2]. The discrepancy in phenotype severity between mutant zebrafish and mutant mice does not seem to be caused by the short isoform in zebrafish predicted in Ensembl; it was not targeted by the gRNAs used in this study because this shorter isoform was not observed in western blot analysis (Supplementary Fig. 2). Thus, the lack of placenta in zebrafish could cause phenotypic differences by preventing embryonic lethality caused by Smg9 deficiency. *smg9*[oi7/oi7] zebrafish live to approximately 6 months of age, enabling a more detailed analysis of the in vivo function of Smg9 in adulthood. Therefore, *smg9*[oi7/oi7] zebrafish allowed us to reveal a link between NMD and premature aging.

Patients with HBMS carrying homozygous mutations in SMG9 display heart, brain, eye, and craniofacial malformation. Our *smg9*[oi7/oi7] zebrafish also exhibited heart and brain abnormalities involving the upregulated expression of *smox* and accumulation of ROS. However, the reason why *SMG9* mutations cause mainly heart and brain defects remains unknown. Our tissue distribution analysis revealed that *smg9* is broadly expressed in various tissues in humans and zebrafish, and *smox* mRNA levels were elevated in the brain, muscle, eye, heart, intestine, and testis of smg9 mutants, suggesting that the oxidative stress associated with increased *smox* expression is not confined to the brain (Supplementary Fig. 12). The tissue specificity affected by the *SMG9* mutation could be due to the greater susceptibility to ROS in the brain and heart.

Determining the link between cardiac and brain abnormalities and the premature aging phenotype is necessary in *smg9*[oi7/oi7] zebrafish[29]. Long-term observation of *smg9*[oi7/oi7] zebrafish treated with Smox and ROS inhibitors during very early developmental stages, as well as *smg9*[oi7/oi7] zebrafish with temporarily restored expression of Smg9 at the larval stage via mRNA

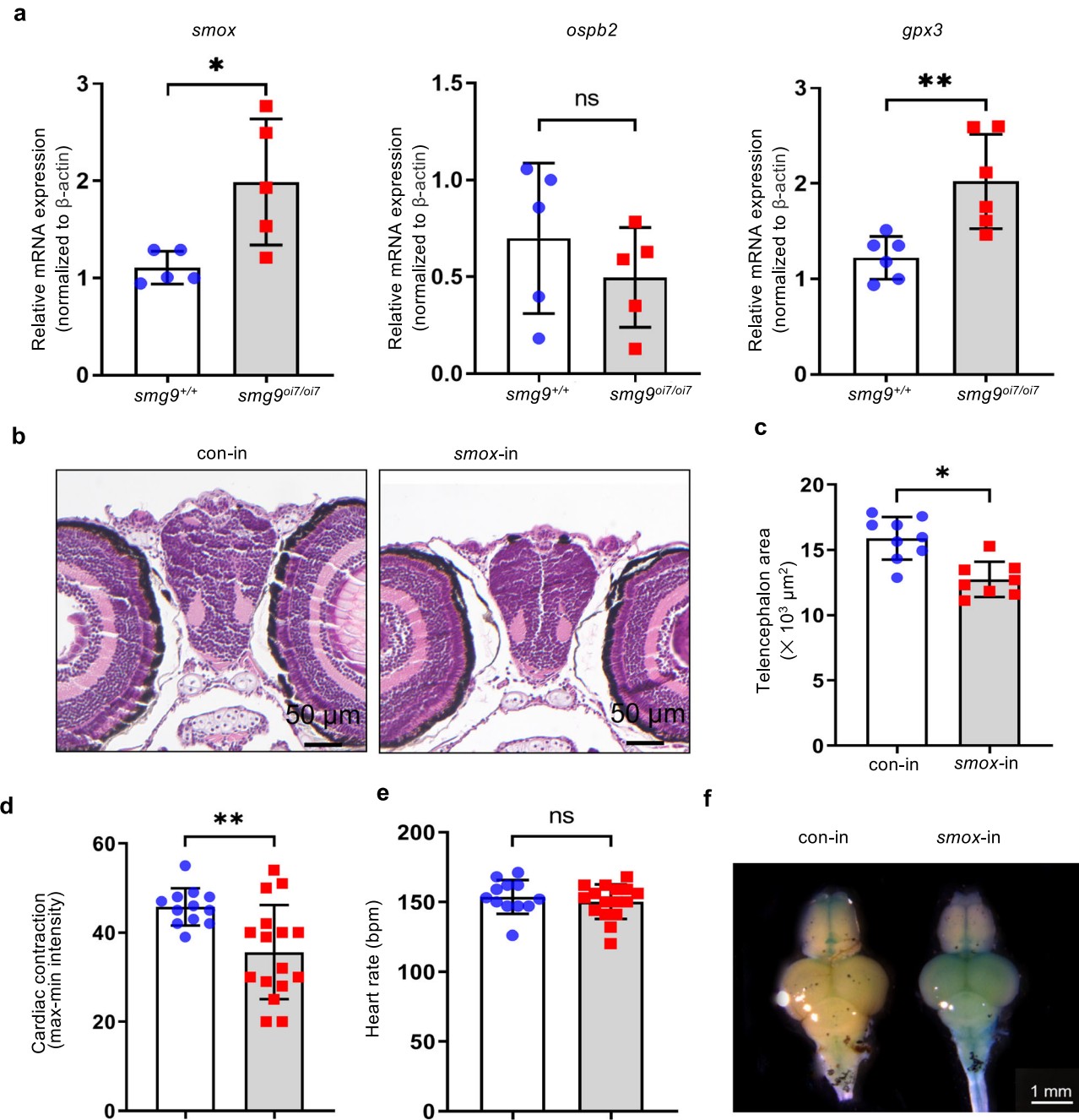

**Fig. 6 | Smox overexpression is sufficient for brain and heart defects in Smg9-deficient zebrafish. a** Quantitative PCR analysis of endogenous target genes of nonsense-mediated mRNA decay in $smg9^{+/+}$ and $smg9^{oi7/oi7}$ larvae at 14 dpf. **b**–**f** Smox mRNA was injected into single-cell-stage fertilized eggs derived from the $smg9^{+/+}$ zebrafish. The telencephalon area at 14 dpf and the heart function at 6 dpf in zebrafish larvae were evaluated. **b** Representative images of H&E staining of smox mRNA-injected (smox-in) and control-injected (con-in) larvae at 14 dpf. Scale bar: 50 μm. **c** Quantification of the telencephalon area of smox-in and con-in larvae at 14 dpf was performed using hematoxylin and eosin staining. **d** Quantification of cardiac contractions in smox-in and con-in larvae at 6 dpf. **e** Quantification of the heart rate of smox-in and con-in larvae at 6 dpf. **f** SA-β-gal staining of the brain of smox-in and con-in zebrafish at 4 mpf. Error bars indicate SD. bpm: beats per minute. *P < 0.05, ** P < 0.01, and *** P < 0.001. ns: not significant.

injection, revealed an improvement in their heart and brain abnormalities during the larval stage. Despite this improvement, these fish exhibited premature aging phenotypes in adulthood (Supplementary Fig. 8 and 16). This result indicates that heart and brain malformation at the developmental stage does not underlie the premature aging phenotypes in $smg9^{oi7/oi7}$ zebrafish. However, further analysis, such as crossbreeding experiments with transgenic zebrafish expressing Smg9 specifically in the brain and heart, will be needed to elucidate the relevance of brain and heart dysfunction in premature aging.

Transcriptome analysis of SMG9 p.(Val184Ala) homozygous patients identified the prevalent upregulation of SMOX, OSBP2, and GPX3 and the downregulation of ZNF115[11]. The orthologs SMOX, OSBP2, and GPX3 were present without ZNF115 in zebrafish. Therefore, we hypothesized that these upregulated genes are involved in the pathogenesis in $smg9^{oi7/oi7}$ zebrafish. We focused on Smox because its upregulation contributes to neuronal damage and aging due to its byproduct, ROS[30]. Our results revealed that Smox overexpression induces developmental abnormalities in the brain, as observed in $smg9^{oi7/oi7}$ zebrafish. We also found that gpx3 expression

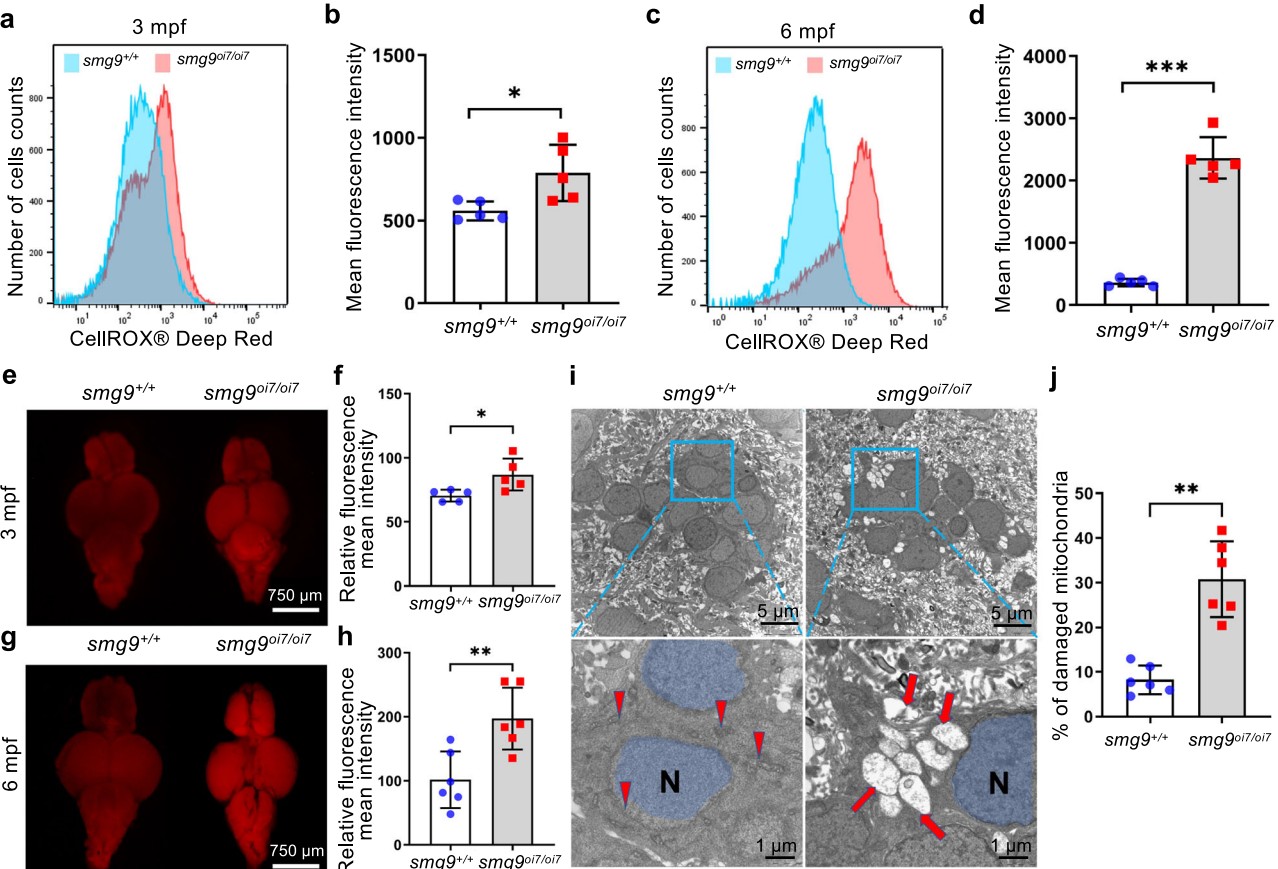

**Fig. 7 | ROS and acrolein accumulation in Smg9-deficient zebrafish.**
**a**, **c** Representative flow cytometry histograms of CellROX fluorescence in $smg9^{+/+}$ and $smg9^{oi7/oi7}$ zebrafish at 3 mpf (**a**) and 6 mpf (**c**). **b**, **d** Mean fluorescence intensity of CellROX Deep Red in the brains of $smg9^{+/+}$ and $smg9^{oi7/oi7}$ zebrafish at 3 mpf (**b**) and 6 mpf (**d**). **e**, **g** Representative images of AcroleinRED staining of $smg9^{+/+}$ and $smg9^{oi7/oi7}$ zebrafish brains at 3 mpf (**e**) and 6 mpf (**g**). **f**, **h** Relative mean fluorescence intensities of $smg9^{+/+}$ and $smg9^{oi7/oi7}$ zebrafish brains at 3 mpf (**f**) and 6 mpf (**h**).

**i**, **j** Representative images of transmission electron microscopy of $smg9^{+/+}$ and $smg9^{oi7/oi7}$ zebrafish brains at 6 mpf. The red arrowhead and red arrow indicate normal and damaged mitochondria, respectively. N Nucleus. Scale bar: 5 μm in the upper panel and 1 μm in the lower panel (**i**). The density of damaged mitochondria within a defined area (0.25 mm²) in the telencephalon region of six individual zebrafish brains (**j**). Error bars indicate SD. *$P < 0.05$, ** $P < 0.01$, and *** $P < 0.001$. ns: not significant.

increased in $smg9^{oi7/oi7}$ zebrafish. Gpx3 acts in opposition to Smox in the regulation of ROS, catalyzing the reduction of hydrogen peroxide by reducing glutathione and protecting cells against oxidative damage[31]. Given that no significant upregulation of $gpx3$ mRNA expression by Smox over-expression was observed in larvae and that Smox inhibitor had no inhibitory effect on $gpx3$ mRNA expression, gpx3 is likely upregulated directly by smg9 deficiency and not downstream of smox. Injection of $gpx3$ mRNA into embryos induces brain hypoplasia, indicating its important role in pheno-typic expression. Other factors whose expression was changed by $smg9$ deficiency may also contribute to HBMS pathogenesis.

NMD is a cellular quality control mechanism that identifies and degrades mRNAs that contain PTCs or stop codons. Endogenous mRNA substrates with several features, such as a uORF, long 3'-UTR, and alter-native splicing isoforms containing a PTC, are also naturally targeted for degradation via the NMD pathway[32]. Because the uORF structure seen in human SMOX mRNA is conserved in zebrafish $smox$ mRNA, the $smg9$ mutation increases $smox$ mRNA by the same mechanism as in patients. Notably, the expression of the endogenous targets of NMD was altered in patients with the SMG9 homozygous missense mutation (Val184Ala), although PTC-containing mRNAs were not affected[11]. The reason behind the absence of changes in PTC-containing mRNA in patients with SMG9 mutations remains unclear. However, the mRNAs that exhibit alterations could include endogenous target genes of NMD distinct from PTC-containing mRNA. RNA-seq-based global transcriptome analysis of

$smg9^{oi7/oi7}$ zebrafish would provide insights into direct or indirect mRNA metabolic systems related to the NMD pathway.

Homozygous mutations in human $SMG8$ cause Alzahrani–Kuwahara syndrome, a neurodevelopmental disorder characterized by dysmorphic facies and cataracts[17]. Although SMG8 deficiency reduces NMD activity in human cells on both endogenous and engineered target transcripts, $smg8$ mutations in $C. elegans$ do not affect NMD targets and phenotypes, indi-cating that $smg8$ is unlikely to encode a component that is critical for NMD in $C. elegans$[33]. Our results showed that $smg8^{oi8/oi8}$ zebrafish had no overt anatomical phenotype and an average lifespan. Further investigations are needed to uncover the in vivo functions of Smg8, including more detailed phenotypic and transcriptomic analyses, particularly of NMD-susceptible transcripts. The interplay between Smg8 and Smg9 in regulating Smg1 activity and Upf1 phosphorylation is an intriguing aspect that warrants further investigation.

Previously, SMG9 knockdown resulted in the augmentation of UPF1 phosphorylation, indicating that SMG9 is essential for suppressing UPF1 phosphorylation[5]. Our cell culture experiments also indicated that the SMG8:SMG9 heterodimer suppresses UPF1 phosphorylation in vitro. This result contradicts the decrease in UPF1 phosphorylation in $smg9^{oi7/oi7}$ zeb-rafish in vivo. The different effects of SMG8/9 on SMG1 activity in vitro and in vivo could be caused by the differences in spatial and temporal regulation. SMG8/9 suppresses the activity of free SMG1 until its recruitment to the EJC, where SMG9 contributes to the stabilization of SMG1 and is also

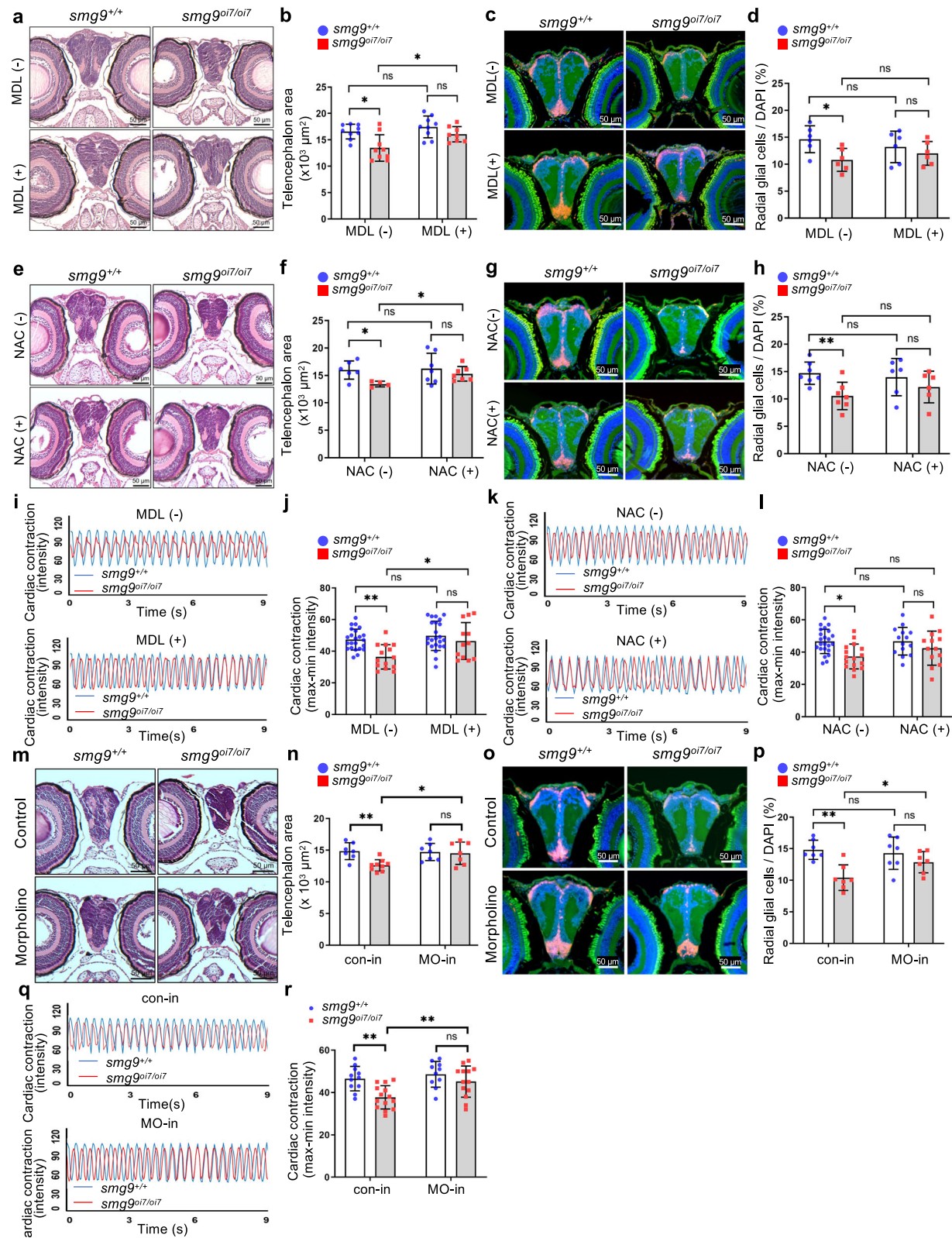

involved in the stability of the efficiency with which SMG8/9 and SMG1 protein complexes are recruited to the EJC[34]. Indeed, reports have shown that NMD is positively regulated by SMG8/9 in vivo, even though SMG8/9 suppresses SMG1 phosphorylation in vitro[5]. Further mechanistic analyses are required to elucidate the regulation of SMG1 activity by SMG8 and SMG9 in vivo.

Our experiments revealed an unexpected mechanistic link between the NMD pathway and premature aging involving SMOX-dependent ROS and acrolein production. Regulation of the SMOX-ROS axis may pave the way for the development of novel therapeutics for HBMS. Additionally, the $smg9^{oi7/oi7}$ zebrafish model established in this study is a valuable tool for investigating the molecular mechanisms

**Fig. 8 | Smox is required for brain and heart defects in Smg9-deficient zebrafish.**
**a** Representative image of hematoxylin and eosin staining of $smg9^{+/+}$ and $smg9^{oi7/oi7}$ larvae after treatment with or without MDL72527. Scale bar: 50 μm. **b** Quantification of the telencephalon area of $smg9^{+/+}$ and $smg9^{oi7/oi7}$ larvae after treatment with or without MDL72527. **c** Representative images of immunostaining for PCNA (red), S100β (green), and DAPI (blue) for $smg9^{+/+}$ and $smg9^{oi7/oi7}$ larvae after treatment with or without MDL72527. Scale bar: 50 μm. **d** Quantification of PCNA/S100β double-positive radial glial cells among DAPI-positive cells in the telencephalon of $smg9^{+/+}$ and $smg9^{oi7/oi7}$ larvae after treatment with or without MDL72527. **e** Representative images of hematoxylin and eosin staining of $smg9^{+/+}$ and $smg9^{oi7/oi7}$ larvae after treatment with or without N-acetylcysteine (NAC). Scale bar: 50 μm. **f** Quantification of the telencephalon area of $smg9^{+/+}$ and $smg9^{oi7/oi7}$ larvae after treatment with or without NAC. Scale bar: 50 μm. **g** Representative images of immunostaining for PCNA (red), S100β (green), and DAPI (blue) for $smg9^{+/+}$ and $smg9^{oi7/oi7}$ larvae after treatment with or without NAC. Scale bar: 50 μm. **h** Quantification of PCNA/S100β double-positive radial glial cells among DAPI-positive cells in the telencephalon of $smg9^{+/+}$ and $smg9^{oi7/oi7}$ larvae, after treatment with or without NAC. **i** Representative dynamic pixel change patterns of the heart showing cardiac contractions of $smg9^{+/+}$ and $smg9^{oi7/oi7}$ zebrafish after treatment with or without MDL72527. **j** Quantification of cardiac contractions in $smg9^{+/+}$ and

$smg9^{oi7/oi7}$ zebrafish larvae after treatment with or without MDL72527.
**k** Representative dynamic pixel change patterns of the heart showing cardiac contractions of $smg9^{+/+}$ and $smg9^{oi7/oi7}$ zebrafish after treatment with or without NAC. **l** Quantification of cardiac contractions in $smg9^{+/+}$ and $smg9^{oi7/oi7}$ zebrafish larvae after treatment with or without NAC. **m** Representative images of hematoxylin and eosin staining of $smg9^{+/+}$ and $smg9^{oi7/oi7}$ larvae injected with control (con-in) or $smox$ morpholino (MO-in). Scale bar: 50 μm. **n** Quantification of the telencephalon area of $smg9^{+/+}$ and $smg9^{oi7/oi7}$ larvae injected with control (con-in) or $smox$ morpholinos (MO-in). **o** Representative images of immunostaining with PCNA (red), S100β (green), and DAPI (blue) for $smg9^{+/+}$ and $smg9^{oi7/oi7}$ larvae injected with control (con-in) or $smox$ morpholino (MO-in). Scale bar: 50 μm. **p** Quantification of PCNA/S100β double-positive radial glial cells among DAPI-positive cells in the telencephalon of $smg9^{+/+}$ and $smg9^{oi7/oi7}$ larvae injected with control (con-in) or $smox$ morpholino (MO-in). **q** Representative dynamic pixel change patterns of the heart showing cardiac contraction of $smg9^{+/+}$ and $smg9^{oi7/oi7}$ zebrafish injected with control (con-in) or $smox$ morpholino (MO-in). **r** Quantification of cardiac contractions in $smg9^{+/+}$ and $smg9^{oi7/oi7}$ zebrafish larvae injected with control (con-in) or $smox$ morpholino (MO-in). Error bars indicate SD. *$P < 0.05$, ** $P < 0.01$, and *** $P < 0.001$. ns: not significant.

---

underlying premature aging and advancing research in the field of aging.

## Methods
### Maintenance of zebrafish and establishment of $smg9^{oi7/oi7}$ and $smg8^{oi8/oi8}$ zebrafish

All experimental animal procedures were performed in accordance with institutional and national guidelines and regulations. The study was conducted in compliance with the ARRIVE guidelines. The study protocol was approved by the Institutional Review Board of Oita University (approval number: 180506). We have complied with all relevant ethical regulations for animal use.

Zebrafish (AB strain; ZFIN, Eugene, OR, USA) were maintained according to standard procedures. Fish were kept under a 10/14 h dark/light cycle at 28 ± 1 °C. The zebrafish were fed twice daily. Embryos were collected and stored at 28.5 °C. Mutant zebrafish lines were generated using the Alt-R CRISPR-Cas9 System (Integrated DNA Technologies, Coralville, IA, USA). Briefly, the guide RNA (gRNA, 5′-CAGCCCGGTCTG-TACGGTCA-3′) for $smg9$ was designed to target the first exon of the $smg9$ locus (Gene ID: 562557) using the CRISPR design tool CRISPRdirect (https://crispr.dbcls.jp), and the gRNA (5′-GACGACACTGA-CAGTCCGGGCGG) was designed for $smg8$ to target the first exon of the $smg8$ locus (gene ID:567424). To synthesize the gRNA complex, crRNA and tracrRNA were mixed to a concentration of 3 mM, heated for 5 min at 95 °C, and cooled to 20–25 °C. Single-cell-stage zebrafish embryos were injected with approximately 1 nL of a solution containing 300 ng/ml Cas9 protein, 30 ng/mL gRNA complex, 20 mM 4-(2-hydroxyethyl)-1-piper-azineethanesulfonic acid, 150 mM KCl, and 0.05% phenol red. The F1 progenies were genotyped using the heteroduplex mobility assay [35] and/or sequencing analysis. Genomic DNA was isolated from tail fin clippings or surface tissue of fish using a live genotyping method[36]. Tissue samples were boiled in 90 mL 50 mM NaOH at 95 °C for 10 min, and then 10 mL 100 mM Tris-HCl (pH 8.0) was added to the buffer as described previously [37]. The primers used for zebrafish genotyping are listed in (Additional file1: Table S1). $smg9^{oi7/+}$ and $smg8^{oi8/+}$ zebrafish were intercrossed to produce $smg9^{oi7/oi7}$ and $smg8^{oi8/oi8}$ homozygous littermates for experimental analyses.

To generate stable transgenic lines of cmlc2-driven enhanced green fluorescent protein (EGFP) [$Tg(cmlc2:EGFP)$][38,39], Tol2-based expression vectors were co-injected with 100 ng/μL Tol2 transposase mRNA into zebrafish embryos at the single-cell stage. EGFP-positive fish were selected to generate founder fish with mosaic expression. $Tg(cmlc2:EGFP)$ zebrafish were crossed with $smg9^{oi7/+}$ zebrafish to obtain Smg9-deficient zebrafish using the $Tg(cmlc2:EGFP)$ transgene. The resulting $smg9^{oi7/+}$ $Tg(cmlc2:EGFP)$ offspring were crossed to generate $smg9^{oi7/oi7}$ $Tg(cmlc2:EGFP)$ and $smg9^{+/+}$

$Tg(cmlc2:EGFP)$ zebrafish as the controls. All adult fish used in this study were male.

### Zebrafish locomotion analysis
Zebrafish behavioral tests were performed using the Pro Stream Webcam C922n (Logicool) to assess the locomotor activity and response to acoustic stimulation in adult $smg9^{+/+}$ and $smg9^{oi7/oi7}$ zebrafish (3 mpf and 6 mpf; $n = 6$/group). A single recording was made for each group to minimize mistracking of individual fish. The fish were allowed to acclimate for 10 min before video recording, and their behavior was recorded for 10 min in a quiet environment. Acoustic stimulation was achieved by tapping the tank five times every 2 min. The swimming behavior of the zebrafish was recorded for 10 min post-stimulation. All data from the locomotion analysis were recorded and analyzed using a Zantiks MWP (Zantiks, Cambridge, UK).

### In vitro mRNA synthesis and microinjection
The coding sequences of zebrafish $smox$ (NCBI reference sequence: NM_001328186.1), zebrafish $gpx3$ (NCBI reference sequence: NM_001137555.3), zebrafish $smg1$ (NCBI Reference Sequence: NM_001080044.2), zebrafish $smg8$ (NCBI Reference Sequence: XM_690719.9), zebrafish $smg9$ (NCBI reference sequence: XM_001923780.7), human SMG1 (NCBI Reference Sequence: XM_047433796.1), human SMG8 (NCBI Reference Sequence: NM_018149.7), human SMG9 (NCBI Reference Sequence: NM_019108.4) and human UPF1 (NCBI reference sequence: NM_001297549.2) were amplified and cloned into a pCS2+ vector containing 3×FLAG. The constructs were linearized and transcribed using the mMESSAGE mMACHI-NETM SP6 Transcription Kit (AM1340; Invitrogen, Carlsbad, CA, USA) according to the manufacturer's protocol. At a concentration of 40 ng/μL, 40 pg of mRNA was injected into each embryo at the single-cell stage using an injection volume of 1 nL. Water without mRNA was injected into single-cell-stage embryos as the negative control. The injected larvae were used for each experiment.

### Analysis of cardiac rhythm and contraction in $Tg(cmlc2:EGFP)$ zebrafish
To evaluate cardiac function, we recorded in vivo videos of the beating hearts of $smg9^{oi7/oi7}$ $Tg(cmlc2:EGFP)$ and $smg9^{+/+}$ $Tg(cmlc2:EGFP)$ larvae at 6 dpf using a fluorescence stereo microscope (Biorevo BZ-9000, KEYENCE, Osaka, Japan). The heart rate and cardiac contractility were measured by analyzing the dynamic pixel change patterns in the EGFP fluorescence signal[40]. Heartbeat videos were captured and imported into the ImageJ software. The EGFP signal was used to outline the heart region to compare the heart size. The circle tool was used to select the same region of interest in

the atrial regions in ImageJ to obtain dynamic pixel changes. The resulting plot profile, which depicts the dynamic pixel changes over time, was obtained for each selected ROI. The obtained data, including peak values and dynamic changes, were exported using ImageJ software. Statistical analysis was subsequently performed to assess differences in cardiac rhythm and contraction between the experimental groups.

## Cryosectioning

The samples were fixed with 4% paraformaldehyde (PFA) for 16 h. Subsequently, they were placed in a microcentrifuge tube containing 30% sucrose in phosphate-buffered saline until the samples fully sank to the bottom. Brain samples were transversally embedded in a mixture of 30% sucrose and Tissue-Tek O.C.T. Compound (4583; Sakura-Finetek, Tokyo, Japan) at a ratio of 2:1 and then fixed on dry ice. Serial sections were obtained using a Leica CM1950 microtome.

## Senescence-associated β-galactosidase (SA-β-gal) staining

Staining was performed as described previously [22]. Briefly, tissues were fixed overnight in 0.2% glutaraldehyde and washed with phosphate-buffered saline (PBS). The tissues were then stained overnight at 37 °C using freshly prepared SA-β-gal (pH 5.9–6.1), consisting of 1 mg/mL 5-bromo-4-chloro-3-indolyl beta-D-galactoside (X-gal, Cell Signaling Technology, Danvers, MA, USA), 40 mM citrate/sodium phosphate (pH 6.0), 5 mM $K_4Fe [CN]_6$, 5 mM $K_3Fe [CN]_6$, 5 mM $K_4Fe [CN]_6$, 2 mM $MgCl_2$, and 150 mM NaCl. After staining, the tissues were washed three times with PBS and photographed using a Leica M205 FA fluorescence stereomicroscope. SA-β-gal signals were quantified using Image J's color threshold selection tool, version 1.52a (National Institutes of Health; Bethesda, MD, USA). A precise thresholding process was employed to define the specific color intensity ranges corresponding to positive SA-β-gal staining while excluding background noise. The lower and upper bounds of the threshold were carefully selected to guarantee precise and repeatable quantification of each sample. SA-β-gal activity was measured by calculating the percentage of positively stained areas relative to the entire area of interest.

## Histopathological examination

The gross appearance of the fish was evaluated under anesthesia using a Leica M205 FA fluorescence stereomicroscope (Leica, Wetzlar, Germany). Body length was measured using the ImageJ software. The brain and testes were collected and observed using a Leica M205 FA fluorescence stereomicroscope. The cerebellum, optic tectum, and telencephalon were analyzed using ImageJ software. After morphological evaluation, tissues were fixed overnight in 4% PFA. After fixation, the tissues were thoroughly washed three times in PBS for 5 min. Subsequently, the tissues were dehydrated using a graded series of ethanol solutions and embedded in paraffin. Embedded tissues were sectioned at a thickness of 5 μm and stained with hematoxylin and eosin (H&E) and Masson trichrome using standard protocols. Stained sections were imaged using an Axio Imager M2 microscope (Carl Zeiss, Jena, Germany).

Immunofluorescence staining was performed according to established protocols [41]. The primary antibodies used were anti-cleaved caspase-3 antibody (D175; Cell Signaling Technology, 1:250), anti-S-100 (IR504; Dako, Glostrup, Denmark), anti-PCNA (M0879; Dako, 1:100), anti-HuC/HuD (A21271; Invitrogen, Carlsbad, CA, USA, 1:100), and anti-phospho-H2A.X (92590, Cell Signaling Technology, 1:250). The secondary antibodies used were Alexa Fluor 488 (A21206; Invitrogen, 1:500) and Alexa Fluor 555 (A21422; Invitrogen; 1:500). After staining, the sections were mounted in fluorescence mounting medium (S3023; Dako) and imaged using an Axio Imager M2 microscope.

## Staining for acrolein in the adult brain

AcroleinRED (FDV-0022, Funakoshi, Tokyo, Japan) staining was performed to visualize acrolein in adult zebrafish brain tissue according to the manufacturer's protocol. Briefly, zebrafish were anesthetized and sacrificed at 3 and 6 mpf, and their brains were removed and incubated in a solution containing 20 mol/L AcroleinRED for 5 min at 20 °C. The samples were then thoroughly washed with PBS, visualized using a Leica M205 FA fluorescent stereo microscope, and quantified using the color threshold selection tool in ImageJ software.

## Measurement of ROS

ROS levels were measured using CellROX (C10444, Invitrogen) with a flow cytometer, according to the manufacturer's instructions. Briefly, brains dissected at 3 and 6 mpf were trypsinized to detach the cells. The collected cells were incubated with CellROX at room temperature for 3 min in the dark and then analyzed for cellular fluorescence intensity. The data were analyzed using the FlowJo software provided by TreeStar (Ashland, OR, USA).

## Transmission electron microscopy

The procedure for electron microscopy analysis of adult zebrafish brains was performed as follows: after anesthetizing the adult zebrafish, the brains were carefully dissected and fixed in 2% glutaraldehyde overnight. The fixed samples were post-fixed with 2% cacodylate-buffered osmium tetroxide, dehydrated in a graded series of ethanol solutions, and embedded in epoxy resin. To prepare the ultrathin sections, resin-embedded tissue blocks were cut using an ultramicrotome. The sections were stained with uranyl acetate and lead citrate and viewed under a transmission electron microscope (H-7650; Hitachi, Tokyo, Japan).

## Western blot analyses in vitro and in vivo

For in vitro experiments, HEK293T cells were transfected with or without the following plasmids: FLAG-human-SMG1 WT, p3xFLAG-cmv10-human-SMG8, p3xFLAG-cmv10-human-SMG9, p3xFLAG-cmv10-zebrafish-Smg1, p3xFLAG-cmv10-zebrafish-Smg8, and p3xFLAG-cmv10-zebrafish-Smg9. FLAG SMG1 WT was a gift from Michael Kastan [42] (Addgene plasmid #199573; http://n2t.net/addgene:199573; RRID: Addgene_199573). The empty vector p3xFLAG-cmv10 was used to standardize the quantity of plasmid DNA for transfection. After 24 h post-transfection, cells were lysed with lysis buffer containing protease and phosphatase inhibitors (Thermo Fisher Scientific, Rockford, IL, USA) on ice for 20 min.

For in vivo experiments, 48 hpf embryos injected with 3xFLAG-UPF1 mRNA were genotyped using a live genotyping method. Twenty larvae from each group were collected in one tube and lysed in a lysis buffer containing protease and phosphatase inhibitors.

The lysates were separated by electrophoresis on a 6–10% sodium dodecyl sulfate-polyacrylamide gel, transferred onto a nitrocellulose membrane (Millipore, Billerica, MA, USA), and probed overnight at 4 °C with primary antibodies against FLAG (mouse anti-FLAG M2 antibody, F1804; Sigma-Aldrich, St. Louis, MO, USA; 1:10000), phospho-UPF1 (Ser1127, Millipore, #07-1016; 1:500), Smg9 (rabbit anti-SMG9 antibody, Bethyl Laboratories, Montgomery, TX, USA, A302-211A; 1:500), β-actin (rabbit anti-β Actin antibody; Abcam, Cambridge, UK, ab8227; 1:10000), and GAPDH (mouse anti-GAPDH antibody; 101M4777 Sigma-Aldrich; 1:2000). The membrane was washed three times with tris-buffered saline/0.1% Tween 20 buffer and incubated with the appropriate secondary antibodies (1:5000 dilution) for 1 h at room temperature. Densitometric analysis was performed using Fusion CAPT Advance Software, version 17.02 (Vilber Lourmat, Collegien, France). Protein bands were quantified using the ImageJ software.

## Reverse-transcription quantitative polymerase chain reaction (RT-qPCR)

For RT-qPCR, six larvae at 3 or 14 dpf were collected in one tube. For the adult zebrafish, individual tissues were dissected and collected separately. Total RNA was extracted using the RNAiso Plus reagent (Takara, Otsu, Japan) according to the manufacturer's instructions. First-strand cDNA was synthesized from 0.5 μg total RNA using ReverTra Ace qPCR RT Master Mix (Toyobo, Osaka, Japan). RT-qPCR was performed using SYBR Green

Power PCR Master Mix (Roche, Mannheim, Germany) on a LightCycler 96 system (Roche) according to the manufacturer's guidelines. Human-SMG9 mRNA expression was analyzed using the human multiple tissue cDNA panel 1 (Clontech, Palo Alto, CA, USA). Primers used for the zebrafish RT-qPCR are listed in Supplementary Table 1.

## Pharmacological treatments

Embryos at 3 hpf ($n = 25$) were treated with 100 μg/mL spermine (Sigma-Aldrich; S3256), 300 μg/mL spermidine (Sigma-Aldrich; S0266), 800 μM MDL72527 (Sigma-Aldrich; M2949), or 50 μm N-acetyl-l-cysteine (Sigma-Aldrich; A7250) for 3 days. The larvae were rinsed with an embryonic medium containing sterile distilled water and transferred to new dishes for further analysis. Larvae at 6 and 14 dpf were used to evaluate heart function and brain phenotype, respectively.

## Morpholino oligonucleotide injection

The *smox*-targeting morpholino antisense oligo (Gene Tools, Philomath, OR, USA; 5′-GATATTTCACAACTTTGCATGCCGT-3′) was prepared by diluting it with water to achieve a working concentration of 0.05 mM, and then 40 pg of *smox* morpholino oligo or control morpholino oligo was microinjected into individual embryos at the single-cell stage.

## Statistics and reproducibility

The results are presented as the mean ± standard deviation (SD). The normality of the data was assessed using the Shapiro–Wilk test for all groups. For comparisons between two groups, Student's *t* test was used for parametric data; the Mann–Whitney U test was applied to non-parametric data. When examining multiple groups, one-way analysis of variance followed by Tukey's post hoc tests was used for parametric data analysis, and the Kruskal–Wallis test with Dunn's post hoc tests was employed for non-parametric data analysis. Mendelian birth incidence of zebrafish genotypes was evaluated using the chi-squared test, with statistical significance set at $P < 0.05$. Statistical analyses were performed using GraphPad Prism software version 8 (GraphPad Software; San Diego, CA, USA). Power analyses were conducted using G*Power software 3.1 (https://www.psychologie.hhu.de/arbeitsgruppen/allgemeine-psychologie-und-arbeitspsychologie/gpower.html) to ensure adequate sample size.

## Reporting summary

Further information on research design is available in the Nature Portfolio Reporting Summary linked to this article.

## Data availability

All data supporting the findings of this study are available within the paper and its Supplementary Information. Sequences of the oligonucleotide primers used in this study are provided in Supplementary Table 1. Any additional data are available from the corresponding author.

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

## Acknowledgements

We thank M. Nakamura-Ota, N. Matsuno, A. Yasuda, K. Miura, H. Wang, and G. Qianyu for their technical assistance in this study. T.H. was supported by the Japan Society for the Promotion of Science [20H03644], the Takeda Science Foundation, the Mizoguchi Urology Clinic, and the Yatsuka Eye Clinic. H.S. was supported by the Japan Society for the Promotion of Science [21K06871].

## Author contributions

S.L. and H.S. conducted most of the experiments and analyzed the data. W.A.S. and N.S. generated the mutant zebrafish. R.U. and R.H. performed locomotor activity analysis. M.I. and K.K. performed histological analysis. T.T. and S.M. performed cardiac contraction analysis. S.Y. and T.S. performed the electron microscopy analysis. A.Y. and T.H. contributed to the study design and discussion of the results. S.L., H.S., and T.H. drafted the manuscript. All the authors have reviewed it.

## Competing interests

The authors declare no competing interests.
