## [Peer Review File · Communications Biology]

Reviewers' comments:

Reviewer #1 (Remarks to the Author):

In this paper, the authors focused on Smg9, one of the Nonsense-mediated mRNA decay (NMD) components, and used its knockout zebrafish to clarify the role of Smg9 in the NMD machinery and proposed a new molecular mechanism for the onset of heart and brain malformation syndrome (HBMS) caused by SMG9 mutations. This paper is highly commendable in that it is the first in zebrafish to clarify the role of Smg9 and Smg8 in the NMD mechanism, which could not be analyzed due to the embryonic lethality of Smg9 knockout mice. The authors first showed that Smg9KO fish exhibit abnormalities in brain development, particularly in the terminal brain, and cardiac contractile function, as well as a short-lived, premature aging phenotype. The molecular mechanism of this phenotype was shown to be due to the accumulation of transcripts of NMD target genes. In contrast, Smg8KO did not exhibit this phenotype, and the addition of Smg8KO to Smg9KO did not alter it, indicating that Smg9 plays an essential role in Upf1 phosphorylation and execution of the NMD machinery. This finding overturns the established notion that Smg9 inhibits the NMD pathway. The authors also hinted at transcriptional variation in human Smg9 mutant patients and showed that the early aging phenotype in Smg9KO is due to accumulation of ROS and acrolein caused by increased Smox expression. This paper also shows that Smox and ROS inhibitors ameliorate defects in brain development and cardiac contractility in Smg9KO fish, paving the way for the treatment of patients with human Smg9 mutations.

I believe that the paper is concise, to the point, and worthy of publication. The reviewer's main concerns are as follows:

- 1) The authors found increased expression of Smox and Gpx3 in Smg9KO, but argue that the former is due to NMD deficiency and the latter is secondary. To clarify this point, (i) Is premature aging not observed in Gpx3-transgenic fish? (ii) Is gpx3 expression enhanced in Smox-transgenic fish? (iii) Does Smox inhibitor reduce Gpx3 expression in Smg9KO? should be investigated.
- 2) The rescue effects of Smox and ROS inhibitors on brain development and cardiac function are very interesting, but it is unfortunate that the analysis is limited to some phenotypes. The quality of this paper would be improved if it could be shown whether each inhibitor also exerts effects on other indicators of premature aging (life span, fertility, motility, and beta-gal activity).
- 3) Are there any additive or synergistic effects between Smox and ROS inhibitors, which may be important in considering whether ROS production in Smg9-KO is solely due to increased Smox expression.

Reviewer #2 (Remarks to the Author):

In the manuscript, "SMG9 links nonsense-mediated mRNA decay to premature aging in zebrafish", the authors report a novel finding that smg9 knockout in zebrafish triggers premature aging and cardiac defects, similar to pathogenic SMG9 mutations in human patients. The manuscript provides convincing evidence of premature aging both on the cellular and organismal levels in smg9 mutants.

Mechanistically, they hypothesize that *smg9* knockout impairs nonsense-mediated mRNA decay (NMD), leading to upregulation of spermidine oxidase (*smox*) and other genes. The byproducts of *Smox*, i.e., reactive oxygen species (ROS) and acrolein, accumulate in the cell and therefore cause oxidative stress, DNA damage, senescence, and, eventually, premature aging.

However, a few major and minor concerns need to be addressed to support these conclusions.

Major concerns

1. The authors introduced a frameshift mutation in *smg9* and *smg8*, respectively, and they predicted these alleles to be null alleles. RT-qPCR and western blot (depending on antibody availability) are necessary to support this hypothesis, or the conclusions need to be tempered.
2. In humans and mice, SMG9 is known to suppress NMD by inhibiting SMG1 kinase activity, and, consequently, UPF1 phosphorylation. In this study, the authors showed that *smg9* knockout in zebrafish leads to upregulation of NMD prone transcripts (Fig. 4a to Fig. 4j) and decreased Upf1 phosphorylation (Fig. 4k and Fig. 4l). Both findings suggest a positive role for *smg9* in NMD. This discrepancy is intriguing. The authors should consider discussing the conservation of *smg9* among different species. More importantly, they should consider looking into Smg1 phosphorylation and/or Smg1 activity to strengthen the link between Smg9, Smg1, Upf1, and NMD. The authors should also explain their choice of using overexpressed human UPF1 instead of endogenous zebrafish Upf1 to assess UPF1 phosphorylation in Fig. 4k and Fig. 4l.
3. The manuscript focuses on *smox* because a previous study has shown that ‘*smox*...were upregulated in patients with the SMG9 homozygous missense mutation (Val184Ala)’ (10). However, this study also showed that SMG9(Val184Ala) does not affect NMD (‘...normal SMG9 function may be involved in transcriptional regulation without affecting nonsense mRNA-induced NMD’). Thus, it is necessary to explain the relevance of *smox* to NMD, e.g., what features render *smox* mRNA an endogenous NMD substrate.
4. In Figure 7, the authors used SMOX or ROS inhibitors to attempt at rescuing the brain and cardiac defects in *smg9* mutants. However, some of these effects are mild and not significantly different between *smg9* mutants with and without inhibitor treatment (Fig. 4c-d, e-f, g-h and k-l). Have the authors tried generating *smox*^{-/-}; *smg9*^{-/-} mutants?
5. Figure 6 clearly showed signs of oxidative stress in the brain of *smg9* mutant. Is this also the case in the heart, which then causes cardiac defect? To establish a causal relationship between *Smox* and the observed phenotypes in different organs, tissue-specific approaches should be taken, such as measuring *smox* expression levels in dissected tissues and genetically manipulate *smox* in a tissue-specific manner.
6. As *smg8* null mutants do not exhibit any phenotype and *smg8* deficiency in *smg9*^{-/-} zebrafish did not affect the phenotype of *smg9*^{-/-}; Are these results suggesting that either Smg8 or Smg9 can suppress the kinase activity of Smg1 until the Smg1-Upf1 complex joins the EJC, whereas only Smg9 plays a role in promoting the phosphorylation of Upf1? Could Smg9 compensate for the loss of Smg8 in the *smg8*^{-/-} mutants (ie, increased transcription or

protein stability) explaining the lack of any phenotype in these mutants (ie, increased NMD or phosphorylation of Upf1)?

7. The brain size of *smg9*^{-/-} zebrafish was restored by exogenous Smg9 supplementation via *smg9* mRNA injection; Does exogenous Smg9 supplementation via *smg9* mRNA injection into *smg9*^{-/-} embryos also restore cardiac contraction and premature ageing?

Minor concerns

1. It would be helpful to clarify the working model of Smox-induced premature aging in *smg9* mutant. Smox catalyzes the conversion of spermine to spermidine, and spermidine 'has been reported to prevent cell senescence, suppress the occurrence or severity of age-related diseases, and prolong lifespan' (Page 13). At the same time, this reaction produces ROS and acrolein, which promote premature aging. The fact that the authors observed elevated ROS and acrolein levels in *smg9* mutant (Fig. 6) but no protective effect of spermine/spermidine treatment (Supplementary Figure 8) suggest that byproducts play a dominant role. Is this the right interpretation?
2. Is there a more quantitative way to measure β -galactosidase activity than imaging (Fig. 3d to Fig. 3k)?
3. The authors should rearrange the panels in Fig. 1 and Fig. 7 so that the panels appear in the right order (a, b, c, d...) in the main text.
4. In Figure 3a, there is an extra space in '(n=2 0)'.
5. In Figure 6a and 6c, both wildtype and *smg9* mutant fish were labeled as *smg9*^{-/-}.

Reviewer #3 (Remarks to the Author):

The authors have created a novel *smg9* zebrafish mutant that is valuable to study as human mutations in Smg9 result in rare disorders affecting the heart and brain. That being said, the zebrafish mutant does not recapitulate the human disease-causing allele(s), and the human allele(s) are only somewhat described and could be better explained. They present a lot of phenotypic data in this manuscript in addition to some molecular analysis that could underly zebrafish mutant deficits. The overexpression of human Upf1 in zebrafish *smg9* mutants is not well controlled for, however, which dampens some enthusiasm for the relevance of this zebrafish model to human disease cases. A lot of the data is well presented in figures, however, I think that throughout some clarification is needed in multiple places and conclusions are incorrectly or overstated for what they are. Lastly, further analysis would strengthen some of these conclusions/results.

Detailed comments:

- The authors should define SMG9 sooner including in the abstract, title, and provide a bit more about its function/ontology as a kinase inhibitor to strengthen the relevance of the manuscript from the start for readers.
- *smg9*^{-/-} should be explained sooner as an indel mutation and a proper zebrafish allele name should be

- created or obtained per zfin.org guidelines. This will keep things consistent in and available to the field.
- where is smg9 expressed? are there any differences in expression between zebrafish and humans that could explain any phenotypic differences?
 - The authors do a good job of documenting advanced aging using molecular markers and other indicators, but could it just be that heart and brain malformations underly this and not smg9 specifically? A long term or timed rescue experiment would strengthen the molecular link to aging for smg9 or an alternate explanation should be provided.
 - The early smox mRNA increase experiment cannot be linked to later aging without making further observations plus use of other molecular tools. Observing acute defects on heart and brain and rescuing these with smox inhibition alone are not enough; observe aging or aging markers later.
 - On both Page 4 results “pathological relationship” and Page 8 “pathological examination” the term pathological is not used correctly.
 - How much smg9 mRNA was used for the rescue? Methods only state that – “Each mRNA (400 ng/μL) was injected into one-cell-stage embryos” How much per embryo or ng per embryo?
 - Remove smg8 mutant info, not needed as there was no phenotype observed yet – focus on smg9 only in this manuscript. It does not strengthen the paper, just makes it longer.
 - Is the human mutant SMG9 protein remaining or do the authors mean a gene mutation in the human gene on page 6 – it is written as a human protein so this is not clear.
 - Page 6, needs revision – “Thus, Smg9 deficiency results in impaired cardiac function but not affect morphogenesis in the developing hearts of zebrafish.” The sentence is not complete.
 - Why is aging and fertility all in supplement figures/data? If aging is a major result as indicated in the title it should be a main figure/main document inclusion and other things removed or moved to the supplement that are less of the main focus.
 - Did the authors test to see if hets (smg9+/-) have shorter life or any notable phenotypes since they have a genotyping protocol?
 - Page 8 – “These data suggest that Smg9 deficiency may lead to premature aging” indicates that the authors lack confidence in these results. “May” is used several times throughout this manuscript, and it would be preferable if they could make more definitive conclusions for more of their experiments.
 - The B-gal results are disorganized and jump between ages and location (skin v. brain) – need to streamline for understandability of the analysis.
 - Many genes including smox need to be defined by name and functionally throughout, not just an abbreviation given.
 - Supp Figure 7 – what age fish?
 - No sense made here, Page 10 – “...resulting in negative regulation of the NMD machinery. Thus, SMG9 deficiency is expected to promote the NMD machinery.” A negative regulation means suppressed, limited, not promoted. This needs to be fixed. Increased target mRNA in zebrafish then means that NMD machinery IS negatively regulated, not promoted; active NMD would downregulate these transcripts instead – why is this result “contrary” ? This is not clear.
 - Could phosphorylation of human UPF1 be different in zebrafish b/c it is fish and not human cells – why not use cell culture to explore this or proper molecular and biochemical controls or co-express human SMG9 with UPF1 in fish? Can human SMG9 rescue smg9-/- to strengthen the human zebrafish connection?
 - Is smg9 targeted by NMD in zebrafish mutants? Need to confirm LOF and that no partly functioning protein is present especially b/c the human mutation is missense (Page 10 described). This is different

than zebrafish created in this study.

- “Notably, ROS production progressively increases with age, consistent with the severity of the disease phenotype.” Page 12 – is progressive really just 2 time points observed? See also Fig 6; Progressive indicates more than two samplings.
- How do you separate heart rhythm versus contraction from a florescence image movie? This is not clear it can be done.
- Regarding stats – are all data normally distributed, any outliers? ANOVA may not be best if not a normal distribution. Is a power test done to detect if all sample sizes are big enough?
- Fig 3 – regarding positive area of the brain, how was this measured, just across the dorsal surface? Are there better images to show this? Tissue sections would be better at equal positions or across an equal number/size of brain sections.
- Were male versus female differences by 3 mos and beyond done to account for any sex differences in smg9 function and NMD?

Reviewer #4 (Remarks to the Author):

Summary:

This manuscript sought to define the role of nonsense-mediated decay (NMD) factor SMG9 in zebrafish. The authors used a CRISPR/Cas9 approach to generate a line of mutant fish, which displayed defects in brain size, cardiac contractility, and induced premature aging phenotypes. They attributed these phenotypes to overexpression of the smox RNA, which the authors show is upregulated in SMG9 mutant fish. The results of the study are interesting and novel, however this reviewer finds a conceptual flaw in the design and description of the smg9^{-/-} as a “null”, as described below. Please address this and the other concerns outlined below.

Major concern:

1. The authors generated a “frameshift deletion” in exon 1, which they state, “likely corresponded to a null allele”. However, the authors fail to experimentally demonstrate this is a null mutation. This is essential to describe the mutants as “null”.

A search of Ensembl by this reviewer found that *Danio rerio* has two isoforms for smg9, which differ by the first two exons (absent in coding for one isoform). A frameshift in exon 1, as the authors have done, likely only disrupts one of the two isoforms and may be a hypomorphic allele rather than a null, which may account for the difference in embryonic lethality compared to the mouse knockouts that the authors reference. The authors must address this issue experimentally (for example, using qRT-PCR or Western blot).

Additional concerns:

1. Authors do not adequately address/discuss differences in phenotypes observed between this study and those in mouse (knockout lethal vs viable) and human (function of SMG8, SMG9 as a positive vs negative regulator of UPF1 phosphorylation).

2. In Introduction, bottom of pg. 2, statement “SMG9 presumably mediates the interaction between SMG8 and SMG1.” needs reference(s).
3. To call smg8 mutant a “null”, needs experimental validation it is a true null (qRT-PCR or Western blot). Otherwise, should just call “mutant” in this case.
4. Including representative time lapse movies of cardiac contraction as depicted in Fig. 2 in still and quantification in Supplementary material would be helpful to illustrate differences in contractility.
5. For statistical analyses, authors must confirm data are normally distributed for parametric tests to be appropriate. Otherwise, nonparametric tests should be used.
6. There is odd spacing in Fig. 3a—(n=2 0) instead of (n=20).
7. In Fig. 3d,f, images look excessively manipulated to remove background around the fish. It is unclear why images weren’t taken similarly to Fig. 3b,c. Also, it is difficult to tell sex of fish in images because the dorsal and anal fins are folded, or possibly due to how the images are manipulated?
8. Scale bars missing in Fig. 3b,c.
9. Description of SA- β -gal signal quantification in Methods is inadequate—please provide better detail regarding how thresholding was performed.
10. Images in Supp Fig 6 need to be better labeled for marker/color, and channels should be separated—it’s difficult to discern differences in HuC/D as displayed.
11. Full blot image in Supplement Fig. 9 should have wells labeled.
12. The authors do not adequately describe/indicate the sex of the adult fish used in the study.

Point-by-point responses to the Reviewers

Thank you very much for reviewing our manuscript. We appreciate the insightful comments of the reviewers. We have provided point-by-point responses to each comment below and made the corresponding revisions to the manuscript document. To make the manuscript revisions easier to locate, they have been highlighted in yellow.

Point-by-point replies to Referee #1

In this paper, the authors focused on Smg9, one of the Nonsense-mediated mRNA decay (NMD) components, and used its knockout zebrafish to clarify the role of Smg9 in the NMD machinery and proposed a new molecular mechanism for the onset of heart and brain malformation syndrome (HBMS) caused by SMG9 mutations. This paper is highly commendable in that it is the first in zebrafish to clarify the role of Smg9 and Smg8 in the NMD mechanism, which could not be analyzed due to the embryonic lethality of Smg9 knockout mice. The authors first showed that Smg9KO fish exhibit abnormalities in brain development, particularly in the terminal brain, and cardiac contractile function, as well as a short-lived, premature aging phenotype. The molecular mechanism of this phenotype was shown to be due to the accumulation of transcripts of NMD target genes. In contrast, Smg8KO did not exhibit this phenotype, and the addition of Smg8KO to Smg9KO did not alter it, indicating that Smg9 plays an essential role in Upf1 phosphorylation and execution of the NMD machinery. This finding overturns the established notion that Smg9 inhibits the NMD pathway. The authors also hinted at transcriptional variation in human Smg9 mutant patients and showed that the early aging phenotype in Smg9KO is due to accumulation of ROS and acrolein caused by increased Smox expression. This paper also shows that Smox and ROS inhibitors ameliorate defects in brain development and cardiac contractility in Smg9KO fish, paving the way for the treatment of patients with human Smg9 mutations.

I believe that the paper is concise, to the point, and worthy of publication. The reviewer's main concerns are as follows:

Response: Thank you for your insightful comments and suggestions. We appreciate the opportunity to address the concerns raised by your review.

1) The authors found increased expression of *Smox* and *Gpx3* in *Smg9KO*, but argue that the former is due to NMD deficiency and the latter is secondary.

To clarify this point,

(i) Is premature aging not observed in *Gpx3*-transgenic fish?

Response: We recognize the importance of investigating premature aging in *Gpx3*-transgenic fish. However, generating *Gpx3*-transgenic fish is time-consuming. To expedite our research and provide relevant insights, we checked the brain size at 14 dpf after injecting *gpx3* mRNA into single-cell-stage embryos. Injection of *gpx3* mRNA significantly decreased brain size (Supplementary Figure 13c, d), suggesting two possibilities: (1) elevated *gpx3* expression contributes to the *smg9* mutant phenotype, as does elevated *smox* expression, or (2) elevated *gpx3* expression occurs downstream of elevated *smox* expression and is responsible for the *smg9* phenotype.

(ii) Is *gpx3* expression enhanced in *Smox*-transgenic fish?

Response: Investigating whether *gpx3* expression is increased in *smox* transgenic fish is reasonable. However, the generation of *smox* transgenic fish is time-consuming. To address this concern in a timely manner, we performed mRNA assessments of *gpx3* at 7 dpf by injecting *smox* mRNA into single-cell-stage embryos. No significant upregulation in *gpx3* mRNA expression was observed (Supplementary Figure 13a). Thus, rather than being downstream of *smox*, *gpx3* seems to be directly regulated by *Smg9* deficiency, similar to *smox*. The manuscript text has been revised, and a description has been added to the Discussion section (page 20).

(iii) Does *Smox* inhibitor reduce *Gpx3* expression in *Smg9KO*? should be investigated.

Response: Investigating the effect of a *Smox* inhibitor on *Gpx3* expression in *smg9^{oi7/oi7}* zebrafish is important. We exposed embryos 3 h post-fertilization to 800 μ M of the SMOX inhibitor MDL72527 for 3 days and subsequently examined *gpx3* mRNA levels. Treatment of *smg9^{oi7/oi7}* zebrafish with *Smox* inhibitors did not significantly affect

gpx3 mRNA levels (Supplementary Figure 13b). Therefore, it is likely that Gpx3 is regulated directly by Smg9 and not downstream of Smox, consistent with the results mentioned in the previous response.

In addition, injection of *gpx3* mRNA into wild-type zebrafish decreased brain size, suggesting that the *smg9^{oi7/oi7}* phenotype is not caused solely by increased *smox* expression but also by increased Gpx3 expression.

2) The rescue effects of Smox and ROS inhibitors on brain development and cardiac function are very interesting, but it is unfortunate that the analysis is limited to some phenotypes. The quality of this paper would be improved if it could be shown whether each inhibitor also exerts effects on other indicators of premature aging (life span, fertility, motility, and beta-gal activity).

Response: We thank you for your constructive feedback and insightful suggestions. To address your comments, we performed additional experiments to evaluate the effects of SMOX and ROS inhibitors on several indicators of premature aging. Embryos were treated at 3 h post-fertilization with 400 μ M MDL72527 and 25 μ M N-acetyl-L-cysteine separately for 14 days. This duration was chosen to minimize the potential toxicity associated with prolonged exposure to inhibitors.

No improvement was observed in premature aging phenotypes (brain SA- β -Gal, survival curve, or fertilization rate) after inhibitor treatment (Supplementary Figure 16). This result may have occurred because treatment with inhibitors only in the juvenile stage does not completely suppress the accumulation of smox and ROS. Generating *smg9* mutant zebrafish with a *smox* deficiency would be necessary to test this hypothesis (page 16).

However, *smox* mRNA injection significantly increased SA- β -Gal levels in 4-month-old brains (Fig. 6f), suggesting that the increase in *smox* mRNA due to Smg9 dysfunction contributes to a premature aging phenotype (page 14).

3) Are there any additive or synergistic effects between Smox and ROS inhibitors, which may be important in considering whether ROS production in Smg9-KO is solely due to increased Smox expression.

Response: We thank the referee for the insightful question regarding the potential additive or synergistic effects of the SMOX and ROS inhibitors. To address this, we treated embryos with 400 μ M MDL72527 and 25 μ M N-acetyl-l-cysteine for 3 days from 3 h post-fertilization and assessed the brain size at 14 dpf (Supplementary Figure 17). The combination of SMOX and ROS inhibitors was effective in suppressing brain phenotypes. However, another factor upregulated by Smg9 deficiency may contribute to ROS accumulation. To resolve this issue, we need to identify the factors related to ROS regulation through comprehensive analysis in the future. We have added this information on page 16. We believe these experimental data improve our understanding of the interplay between Smox and ROS inhibitors in ameliorating the phenotypes observed in the Smg9 mutant zebrafish.

Point-by-point replies to Referee #2

In the manuscript, "SMG9 links nonsense-mediated mRNA decay to premature aging in zebrafish", the authors report a novel finding that smg9 knockout in zebrafish triggers premature aging and cardiac defects, similar to pathogenic SMG9 mutations in human patients. The manuscript provides convincing evidence of premature aging both on the cellular and organismal levels in smg9 mutants. Mechanistically, they hypothesize that smg9 knockout impairs nonsense-mediated mRNA decay (NMD), leading to upregulation of spermidine oxidase (smox) and other genes. The byproducts of Smox, i.e., reactive oxygen species (ROS) and acrolein, accumulate in the cell and therefore cause oxidative stress, DNA damage, senescence, and, eventually, premature aging.

However, a few major and minor concerns need to be addressed to support these conclusions.

Major concerns

1. The authors introduced a frameshift mutation in smg9 and smg8, respectively, and they predicted these alleles to be null alleles. RT-qPCR and western blot (depending on antibody availability) are necessary to support this hypothesis, or the conclusions need to be tempered.

Response: We thank you for your thoughtful evaluation of our manuscript and suggestion to provide additional support for the characterization of the frameshift mutations in *smg9* and *smg8* as null alleles.

The anti-human SMG9 polyclonal antibody from Bethyl Laboratories reacted with the zebrafish Smg9. Therefore, we performed western blot analysis and confirmed the absence of Smg9 protein expression (Fig. 1b). A minor isoform reported in Ensembl was not observed, indicating that there was no expression of this protein isoform in zebrafish (**Supplementary Figure 2a**). We registered this *smg9* mutant zebrafish with ZFIN as *smg9^{oi7/oi7}* and refer to it as such in the revised manuscript (**pages 4–5**).

We were unable to obtain an antibody that reacted with zebrafish Smg8. Therefore,

RT-qPCR was performed to confirm the deletion of the wild-type *smg8* mRNA (Supplementary Figure 3c). Because the possibility of hypomorphism cannot be ruled out for *smg8*^{-/-}, referring to it as “mutant” rather than “null” is appropriate. It was registered in ZFIN as *smg8*^{oi8/oi8} and is referred to as such in the revised manuscript (page 6).

2. In humans and mice, SMG9 is known to suppress NMD by inhibiting SMG1 kinase activity, and, consequently, UPF1 phosphorylation. In this study, the authors showed that smg9 knockout in zebrafish leads to upregulation of NMD prone transcripts (Fig. 4a to Fig. 4j) and decreased Upf1 phosphorylation (Fig. 4k and Fig. 4l). Both findings suggest a positive role for smg9 in NMD. This discrepancy is intriguing.

(i) The authors should consider discussing the conservation of smg9 among different species.

Response: SURF components are highly conserved in vertebrates (Causier et al., *Scientific Rep* 2017), with more than 70% of the amino acids conserved between zebrafish and humans. Structural analysis of the human SMG1 protein revealed that the domain of SMG important for regulating activity is conserved. We have added this information on page 12–13.

(ii) More importantly, they should consider looking into Smg1 phosphorylation and/or Smg1 activity to strengthen the link between Smg9, Smg1, Upf1, and NMD.

Response: SMG1 activity correlates with UPF1 phosphorylation, with increased UPF1 phosphorylation resulting in increased SMG1 activity. Therefore, SMG1 activity was tested by measuring the degree of UPF1 phosphorylation.

To address the referee’s suggestion, we examined whether human and zebrafish SMG8 and SMG9 are interchangeable in the SMG1:SMG8:SMG9 complex. First, we overexpressed the human SMG8:SMG9 heterodimers in 293T cells. As previously reported (Yamashita et al., *Genes Dev* 2009), the human SMG8:SMG9 heterodimer suppresses SMG1 activity in vitro. This effect was also observed upon overexpression of

the zebrafish Smg8:Smg9 heterodimer. Furthermore, human SMG8 and zebrafish SMG9 heterodimers, and conversely, zebrafish SMG8 and human SMG9 heterodimers, inhibited the activity of human SMG1 in UPF1 phosphorylation in vitro (Supplementary Figure 10b). These results indicate that the components of the SMG1:SMG8:SMG9 complex are mutually interchangeable between humans and zebrafish. However, this result conflicts with the decrease in UPF1 phosphorylation in *smg9^{oi7/oi7}* zebrafish observed in vivo (pages 12–13).

The different effects of SMG8/9 on SMG1 activity in vitro and in vivo may be caused by differences in spatial and temporal regulation. SMG9 is involved in the stability of SMG1 and the efficiency with which the SMG8/9 and SMG1 protein complex is recruited to EJC (Arias-Palomo et al., *Genes Dev* 2011). Indeed, previous reports have shown that NMD is regulated positively by SMG8/9 in vivo, even though SMG8/9 suppresses SMG1 phosphorylation in vitro (page 22).

(iii) The authors should also explain their choice of using overexpressed human UPF1 instead of endogenous zebrafish Upf1 to assess UPF1 phosphorylation in Fig. 4k and Fig. 4l.

Response: We injected human UPF1 instead of zebrafish Upf1 because we had no antibodies that recognized phosphorylated UPF1 in zebrafish. We confirmed that the anti-human phosphorylated UPF1 antibody did not recognize phosphorylated Upf1 in 293T cells transfected with zebrafish Smg1 and Upf1. We used human 293T cells to test whether zebrafish Smg1 can phosphorylate human UPF1 and found that the phosphorylation of UPF1 in 293T cells was enhanced by zebrafish Smg1 expression (Supplementary Figure 10a). An explanation of this result is provided in the text. An explanation has been added on pages 12–13.

3. The manuscript focuses on smox because a previous study has shown that ‘smox...were upregulated in patients with the SMG9 homozygous missense mutation (Val184Ala)’ (10). However, this study also showed that SMG9(Val184Ala) does not affect NMD (‘...normal SMG9 function may be involved in transcriptional regulation without affecting nonsense

mRNA-induced NMD'). Thus, it is necessary to explain the relevance of smox to NMD, e.g., what features render smox mRNA an endogenous NMD substrate.

Response: Thank you for your suggestion to clarify the relevance of smox in nonsense-mediated mRNA decay (NMD). This cellular quality-control mechanism identifies and degrades mRNAs containing premature termination codons (PTCs). In NMD, endogenous mRNA substrates are naturally targeted for degradation. Several features in addition to PTCs, such as a uORF, long 3'-UTR, and alternative splicing–produced PTC, can make mRNA an NMD substrate (Peccarelli and Kebaara, *Eukaryot Cell* 2014).

The mRNAs containing a uORF or long 3'-UTR are thought to be regulated by the same machinery that recognizes PTC-containing mRNA; however, Rahikkala et al. reported that endogenous NMD targets are modified in peripheral blood samples in patients carrying *SMG9* homozygous missense mutation, although PTC-containing mRNAs were not affected. The uORF structure in human *SMOX* mRNA is conserved in zebrafish *smox* mRNA, suggesting the *smg9* mutation increases *smox* mRNA by the same mechanism as in patients.

We did not perform a comprehensive expression analysis of the changes in the expression of PTC-containing alleles in *Smg9^{oi7/oi7}*; however, PTC-containing mRNAs might not be affected in *Smg9^{oi7/oi7}* as seen in human patients. The reason for the absence of changes in PTC-containing mRNA in patients with SMG9 mutations remains unclear. The relatively mild decrease in NMD activity caused by the hypomorphic missense mutation in human SMG9 may have a greater effect on these endogenous targets. We have added this information to the Discussion (pages 20–21).

4. *In Figure 7, the authors used SMOX or ROS inhibitors to attempt at rescuing the brain and cardiac defects in smg9 mutants. However, some of these effects are mild and not significantly different between smg9 mutants with and without inhibitor treatment (Fig. 4c-d, e-f, g-h and k-l). Have the authors tried generating smox-/-; smg9-/- mutants?*

Response: For those that were not statistically significantly different with and without inhibitor treatment, we have revised the text to read “tendency to be restored” (page 16).

We agree that it would be best to create and validate *smox*^{-/-} and *smg9*^{-/-} double mutants, as suggested by the referee. However, considering the time constraints associated with generating double mutants of *smox*^{-/-} and *smg9*^{oi7/oi7}, we adopted an alternative approach employing *smox* morpholino knockdown. As shown in Figure 8m-r, significant phenotypic improvement was observed with the inhibitor treatment. In addition, more defined improvements were observed in simultaneous treatment with *Smox* inhibitors and NAC (Supplementary Figure 17) (page 16).

5. Figure 6 clearly showed signs of oxidative stress in the brain of smg9 mutant. Is this also the case in the heart, which then causes cardiac defect? To establish a causal relationship between Smox and the observed phenotypes in different organs, tissue-specific approaches should be taken, such as measuring smox expression levels in dissected tissues and genetically manipulate smox in a tissue-specific manner.

Response: We appreciate the reviewer's insightful comment and the importance of investigating the potential causal relationship between increased *Smox* expression, oxidative stress, and the observed cardiac defects in the *smg9* mutant.

In response to the referee's suggestion, we measured the expression levels of *Smox* in various *smg9*^{-/-} zebrafish tissues. Our analysis revealed that elevated *smox* mRNA levels were observed in the brain, muscle, eye, heart, intestine, and testes of *smg9* mutants, suggesting that the oxidative stress associated with increased *Smox* expression is not confined to the brain (Supplementary Figure 12). The brains and hearts of larvae exhibit particularly noticeable phenotypes indicative of their greater susceptibility to ROS. We have added this discussion (page 18–19).

This observation strongly suggested a potential association between elevated *Smox* expression and oxidative stress in cardiac tissues, providing a plausible mechanism for the observed cardiac defects. To further explore the tissue-specific effects of *Smox* and its role in cardiac defects, we need to conduct tissue-specific genetic manipulations of *Smox*, as suggested by the referee. This approach will provide valuable insights into the direct effect of *Smox* on cardiac function and help establish a causal relationship between *Smox*, oxidative stress, and the phenotypes observed in different organs.

6. As *smg8* null mutants do not exhibit any phenotype and *smg8* deficiency in *smg9*^{-/-} zebrafish did not affect the phenotype of *smg9*^{-/-}; Are these results suggesting that either *Smg8* or *Smg9* can suppress the kinase activity of *Smg1* until the *Smg1-Upf1* complex joins the EJC, whereas only *Smg9* plays a role in promoting the phosphorylation of *Upf1*? Could *Smg9* compensate for the loss of *Smg8* in the *smg8*^{-/-} mutants (ie, increased transcription or protein stability) explaining the lack of any phenotype in these mutants (ie, increased NMD or phosphorylation of *Upf1*)?

Response: No difference in the phenotype being observed between *smg9*^{-/-} and *smg8/9* double-KO is reasonable because a biochemical and structural study revealed that SMG8 binds to SMG1 via SMG9 (Fernandez et al., *Nucleic Acids Res* 2011). However, the lack of a phenotype in *smg8*^{-/-} zebrafish was surprising.

As the referee proposed, *Smg9* might compensate for *Smg8* deficiency because the NMD pathway is characterized by an autoregulatory negative feedback mechanism that regulates the expression of core NMD factors through the NMD machinery (Huang et al., *Mol Cell* 2011). Therefore, we examined *smg9* mRNA levels in *smg8*^{-/-} zebrafish, finding a notable decrease in *smg9* mRNA levels compared with that in wild type (Supplementary Figure 5), suggesting that *Smg9* does not compensate for a *Smg8* deficiency.

The interaction between *Smg8* and *Smg9* in regulating *Smg1* activity and *Upf1* phosphorylation is an intriguing aspect that warrants further investigation. We understand that SMG8/9 suppresses the activity of free SMG1 until its recruitment to the EJC, where SMG9 contributes to the stabilization of SMG1 protein, similar to the regulation of p110 of the same family, PI3K, by p85 (Fox et al., *Biochem Soc Trans* 2020).

Although SMG1 activity is regulated mainly by SMG8, SMG9 might contribute more to SMG1 stabilization than SMG8 (Arias-Palomo et al., *Genes Dev* 2011). After recruitment to the EJC, it is possible that SMG9, rather than SMG8, positively regulates the activity of SMG1. SMG9 alone does not suppress SMG1 kinase activity in vitro (Deniaud et al., *Nucleic Acids Res* 2015). Indeed, we observed that overexpression of SMG9 alone in HEK293T cells did not suppress UPF1 phosphorylation, unlike the co-overexpression of SMG8/9 (Supplementary Figure 10b and data not shown).

In addition, PTC-containing mRNAs and other endogenous targets of NMD could be regulated by different mechanisms of the NMD pathway. SMG8 may be required for regulating PTC-containing mRNA, but SMG9 may contribute more to regulating the other endogenous targets of NMD. However, this requires clarification in future studies.

These factors may explain the lack of a phenotype in *smg8* mutants. We have added this information to the Discussion section (pages 21–22).

7. The brain size of smg9–/– zebrafish was restored by exogenous Smg9 supplementation via smg9 mRNA injection; Does exogenous Smg9 supplementation via smg9 mRNA injection into smg9–/– embryos also restore cardiac contraction and premature ageing?

Response: We appreciate the referee’s interest in understanding the broader effects of exogenous Smg9 supplementation on cardiac contraction and premature aging in *smg9^{oi7/oi7}* zebrafish. To address this, we injected zebrafish *smg9* mRNA into single-cell-stage embryos, evaluating cardiac contractions at six days post-fertilization and premature aging phenotypes in adult fish (Fig. 2g and **Supplementary Figure 8**). Although exogenous Smg9 supplementation via *smg9* mRNA injection into *smg9^{oi7/oi7}* embryos restored the brain and heart phenotypes at the larval stage (Fig. 2g) (**page 7**), it did not ameliorate premature aging phenotypes such as lifespan, fertility, and β -galactosidase activity (**Supplementary Figure 8**). Because the premature aging phenotype in *smg9^{oi7/oi7}* zebrafish is observed at the adult stage, long-term expression of exogenous Smg9 is required to elucidate the effect of exogenous Smg9 expression on premature aging phenotypes. The results of these experiments are presented in the revised manuscript (**page 10–11**).

Minor concerns

1. It would be helpful to clarify the working model of Smox-induced premature aging in smg9 mutant. Smox catalyzes the conversion of spermine to spermidine, and spermidine ‘has been reported to prevent cell senescence, suppress the occurrence or severity of age-related diseases, and prolong lifespan’ (Page 13). At the same time, this reaction

produces ROS and acrolein, which promote premature aging. The fact that the authors observed elevated ROS and acrolein levels in smg9 mutant (Fig. 6) but no protective effect of spermine/spermidine treatment (Supplementary Figure 8) suggest that byproducts play a dominant role. Is this the right interpretation?

Response: We appreciate the reviewer's insightful observation. To make it more understandable for the reader, we have modified the text as follows (pages 15–16).

Smox catalyzes the conversion of spermine to spermidine, and spermidine prevents cell senescence, suppresses the occurrence or severity of age-related diseases, and prolongs lifespan (Al-Habsi et al., *Science* 2022, Ni & Liu, *Aging Dis* 2021). At the same time, this reaction produces ROS and acrolein, which promote premature aging. The smg9 mutant displayed high ROS and acrolein levels (Fig. 7). However, no protective effects of spermine/spermidine treatment were observed (Supplementary Figure 15). These results suggest that byproducts play a dominant role in the premature aging phenotype.

2. Is there a more quantitative way to measure β -galactosidase activity than imaging (Fig. 3d to Fig. 3k)?

Response: While our current approach relies on imaging for qualitative assessment, we recognize the importance of incorporating a quantitative method for comprehensive analysis. To improve the quantitation of β -galactosidase activity, frozen sections were used for brain and β -galactosidase staining, providing a more accurate representation of the positive brain areas (Fig. 4g-l and Supplementary Figure 6)

3. The authors should rearrange the panels in Fig. 1 and Fig. 7 so that the panels appear in the right order (a, b, c, d...) in the main text.

Response: We apologize for the inadequacies in the figure panels. We have corrected them.

4. In Figure 3a, there is an extra space in '(n=2 0)'.

Response: We have corrected the sentence accordingly.

5. *In Figure 6a and 6c, both wildtype and smg9 mutant fish were labeled as smg9-/-.*

Response: We thank the referee for pointing this out. We have corrected this error.

Point-by-point replies to Referee #3

The authors have created a novel smg9 zebrafish mutant that is valuable to study as human mutations in Smg9 result in rare disorders affecting the heart and brain. That being said, the zebrafish mutant does not recapitulate the human disease-causing allele(s), and the human allele(s) are only somewhat described and could be better explained. They present a lot of phenotypic data in this manuscript in addition to some molecular analysis that could underly zebrafish mutant deficits. The overexpression of human Upf1 in zebrafish smg9 mutants is not well controlled for, however, which dampens some enthusiasm for the relevance of this zebrafish model to human disease cases. A lot of the data is well presented in figures, however, I think that throughout some clarification is needed in multiple places and conclusions are incorrectly or overstated for what they are. Lastly, further analysis would strengthen some of these conclusions/results.

Detailed comments:

1) The authors should define SMG9 sooner including in the abstract, title, and provide a bit more about its function/ontology as a kinase inhibitor to strengthen the relevance of the manuscript from the start for readers.

Response: We appreciate the referee's thoughtful comments. We have revised the text so that the reader has a better understanding of SMG 9 as follows:

Title:

Effect of nonsense-mediated mRNA decay factor SMG9 deficiency on premature aging in zebrafish

Abstract:

SMG9 is an essential component of the nonsense-mediated mRNA decay (NMD) machinery, a quality control mechanism that selectively degrades aberrant transcripts. Mutations in SMG9 are associated with heart and brain malformation syndrome (HBMS). However, the molecular mechanism underlying HBMS remains unclear.

2) *smg9*^{-/-} should be explained sooner as an indel mutation and a proper zebrafish allele name should be created or obtained per zfin.org guidelines. This will keep things consistent in and available to the field.

Response: We have registered our *smg9* and *smg8* mutant zebrafish in ZFIN as *smg9*^{oi7/oi7} and *smg8*^{oi8/oi8}, respectively, and refer to them as such in the revised manuscript (pages 4, 6).

3) where is *smg9* expressed? are there any differences in expression between zebrafish and humans that could explain any phenotypic differences?

Response: We appreciate the referee's interest in the expression of *smg9*. To address this, we analyzed *smg9* mRNA expression in various zebrafish tissues at 3 months of age and compared it with human tissues (Supplementary Figure 1). We hope this supplement will provide a more comprehensive understanding of *smg9* expression in different species. Except for the testes, the SMG9 mRNA expression patterns were approximately the same, suggesting the phenotypic differences between zebrafish and humans are not due to differences in gene expression. Given that mutations in human patients result in a missense mutation rather than a null mutation, as in our *smg9*^{oi7/oi7} model, a residual hypomorphic function may still be present (page 4).

4) The authors do a good job of documenting advanced aging using molecular markers and other indicators, but could it just be that heart and brain malformations underly this and not *smg9* specifically? A long term or timed rescue experiment would strengthen the

molecular link to aging for smg9 or an alternate explanation should be provided.

Response: We agree with your suggestion. Although SMOX and ROS inhibitor treatment ameliorated the brain and heart phenotypes in *smg9^{oi7/oi7}* larvae, it did not ameliorate premature aging phenotypes, such as lifespan, fertility, and β -galactosidase activity (Supplementary Figure 16). Timed rescue of Smg9 expression measured by injecting *smg9* mRNA into single-cell embryos also did not rescue the premature aging phenotype in *smg9^{oi7/oi7}* zebrafish (Supplementary Figure 8). Thus, heart and brain malformations at the developmental stage do not seem to underlie premature aging phenotypes in adults. Crossbreeding experiments with transgenic zebrafish expressing Smg9, specifically in the brain and heart, will help elucidate the relevance of brain and heart dysfunction in premature aging (pages 10–11 and 16,19).

5) The early smox mRNA increase experiment cannot be linked to later aging without making further observations plus use of other molecular tools. Observing acute defects on heart and brain and rescuing these with smox inhibition alone are not enough; observe aging or aging markers later.

Response: To address this concern, we performed an additional experiment in which zebrafish *smox* mRNA was injected into single-cell-stage embryos. We observed and assessed aging markers, including lifespan and fertility, in adult fish to gain a better understanding of the long-term effects of Smox modulation (Fig. 6f, Supplementary Figure 14) (page 14). Because overexpression of Smox during only the larval stage increases SA- β -gal staining in the brains of adult fish, accumulation of ROS during the larval stage should be sufficient to induce an early aging phenotype (Fig. 6f) (page 14).

We hope this additional analysis clarifies the link between early molecular changes and later-aging outcomes in our study. However, the lifespan and fertility of *smox* mRNA-injected zebrafish were similar to those of the control group, indicating that short-term smox expression is insufficient to induce full aging phenotypes. Investigating whether Smox upregulation is required for the premature aging phenotypes in *smg9^{oi7/oi7}* zebrafish has an experimental limitation because long-term treatment with SMOX and ROS inhibitors is impossible due to the toxicity of these inhibitors. Generating *smg9^{oi7/oi7}*

zebrafish with a smox deficiency would be useful for testing this hypothesis.

6) On both Page 4 results “pathological relationship” and Page 8 “pathological examination” the term pathological is not used correctly.

Response: Regarding the term “pathological” on pages 4 and 9, we have changed the text as follows: “To clarify the pathophysiological relationship between HBMS and SMG9, we generated *smg9* mutant zebrafish using CRISPR/Cas9 genome editing” (page 4). On page 9, the text was revised to “Histological examination of the testes revealed a decreased number of mature sperm cells in *smg9^{oi7/oi7}* zebrafish.”

7) How much *smg9* mRNA was used for the rescue? Methods only state that – “Each mRNA (400 ng/μL) was injected into one-cell-stage embryos” How much per embryo or ng per embryo?

Response: We apologize for the inadequate description in the Methods section. We have added the following statement (page 26):

Forty picograms of each mRNA were injected into individual embryos at the single-cell stage.

8) Remove *smg8* mutant info, not needed as there was no phenotype observed yet – focus on *smg9* only in this manuscript. It does not strengthen the paper, just makes it longer.

Response: We appreciate the reviewer’s suggestion. However, following the editor’s guidance, we have retained the *smg8* mutant data.

9) Is the human mutant SMG9 protein remaining or do the authors mean a gene mutation in the human gene on page 6 – it is written as a human protein so this is not clear.

Response: We apologize for the errors in the description. We have changed SMG9 to SMG9 (page 7).

10) Page 6, needs revision – “Thus, *Smg9* deficiency results in impaired cardiac function but not affect morphogenesis in the developing hearts of zebrafish.” The sentence is not complete.

Response: We apologize for the grammatical error. We have corrected it as follows (page 7):

Thus, *Smg9* deficiency results in impaired cardiac function but does not affect morphogenesis in the developing hearts of zebrafish.

11) Why is aging and fertility all in supplement figures/data? If aging is a major result as indicated in the title it should be a main figure/main document inclusion and other things removed or moved to the supplement that are less of the main focus.

Response: We thank you for your comment regarding figure organization and the importance of presenting aging and fertility data more prominently in the main text. The figures have been restructured, and the aging phenotype of the testes is now presented in Figure 3.

12) Did the authors test to see if hets (*smg9*^{+/-}) have shorter life or any notable phenotypes since they have a genotyping protocol?

Response: We appreciate the referee’s suggestion to explore the potential phenotypes of heterozygous *smg9* mutant zebrafish. Our analyses did not reveal any notable phenotypes or differences in lifespan compared with their wild-type counterparts. The

survival rate data for heterozygous zebrafish are shown in Figure 3a and described on page 8.

13) Page 8 – “These data suggest that *Smg9* deficiency may lead to premature aging” indicates that the authors lack confidence in these results. “May” is used several times throughout this manuscript, and it would be preferable if they could make more definitive conclusions for more of their experiments.

Response: Regarding using the word ‘may’ in our manuscript, we understand the importance of clarity and confidence in scientific statements. In addition, the valuable suggestions of the four referees have given us more confidence in our results. Therefore, the word ‘may’ in the revised manuscript is now used only twice (pages 20 and 23).

14) The *B-gal* results are disorganized and jump between ages and location (skin v. brain) – need to streamline for understandability of the analysis.

Response: We have presented the results of β -galactosidase staining based on age (1, 3, and 6 months) while maintaining a clear separation between the data for skin and brain tissues (Fig. 4a-1) (pages 9–10).

15) Many genes including *smox* need to be defined by name and functionally throughout, not just an abbreviation given.

Response: We have carefully considered this suggestion to define gene names containing “smox” both by their full name and functionally and have added this information to Supplementary Table 2. We have added the exact names of these abbreviations on pages 11, 13.

16) *Supp Figure 7* – what age fish?

Response: We apologize for the lack of information on age. In Supplementary Figure 7, we used zebrafish at six months post-fertilization, as indicated in the figure legend.

17) No sense made here, Page 10 – “...resulting in negative regulation of the NMD machinery. Thus, SMG9 deficiency is expected to promote the NMD machinery.” A negative regulation means suppressed, limited, not promoted. This needs to be fixed. Increased target mRNA in zebrafish then means that NMD machinery IS negatively regulated, not promoted; active NMD would downregulate these transcripts instead – why is this result “contrary” ? This is not clear.

Response: We apologize for the ambiguity and misinterpretation of this description. This sentence was removed, and the following sentence was added (page 12):

Phosphorylated UPF1 promotes the NMD pathway. To assess the effect of SMG9 deficiency on UPF1 phosphorylation in vivo, we expressed the exogenous human UPF1 protein in *smg9^{oi7/oi7}* larvae and examined its phosphorylation status.

18) Could phosphorylation of human UPF1 be different in zebrafish b/c it is fish and not human cells – why not use cell culture to explore this or proper molecular and biochemical controls or co-express human SMG9 with UPF1 in fish? Can human SMG9 rescue *smg9*^{-/-} to strengthen the human zebrafish connection?

Response: We injected human UPF1 instead of zebrafish Upf1 because we had no antibodies that recognized phosphorylated zebrafish Upf1. We confirmed that human UPF1 is a substrate for zebrafish Smg1 and that phosphorylation of human UPF1 was enhanced by zebrafish Smg1 in 293T cells (Supplementary Fig. 10a). An explanation has been added on pages 12–13.

Based on the referee’s suggestions, we used cell culture to explore the effect of human and zebrafish SMG family molecules on UPF1 phosphorylation by SMG1 (Supplementary Figure 10b). The results indicate that the components of the SMG1:SMG8:SMG9 complex are interchangeable between humans and zebrafish (page 12–13). Moreover, zebrafish *smg9* and human *smg9* are functionally equivalent because the phenotype of *smg9* mutants is improved by the injection of human *smg9* mRNA (Supplementary Figure 11).

19) Is *smg9* targeted by NMD in zebrafish mutants? Need to confirm LOF and that no partly functioning protein is present especially b/c the human mutation is missense (Page 10 described). This is different than zebrafish created in this study.

Response: Whether *smg9* is a target of NMD in *smg9^{oi7/oi7}* was tested by RT-qPCR analysis using heterozygous *smg9* mutant fish because NMD does not function properly in *smg9* homozygous mutant zebrafish. In heterozygotes, mutated *smg9* mRNA levels were significantly reduced, suggesting that *smg9* mutant mRNA is a target of NMD.

We agree that confirming whether the mutation in this zebrafish model is loss-of-function or hypomorphic is important. We performed western blot analysis and confirmed the absence of Smg9 protein expression (Fig. 1b). A minor isoform reported in Ensembl was not observed, indicating that the protein isoform had no expression in zebrafish (Supplementary Fig. 2a) (page 5). Therefore, it is considered to be a loss of function.

Because our *smg9^{oi7/oi7}* zebrafish model exhibits phenotypes in the heart and brain, we believe it mimics the human *SMG9* mutant phenotype to some extent. To test this, generating a knock-in zebrafish with the same missense mutation would be useful; this will be the subject of future work.

20) “Notably, ROS production progressively increases with age, consistent with the severity of the disease phenotype.” Page 12 – is progressive really just 2 time points observed? See also Fig 6; Progressive indicates more than two samplings.

Response: We have removed the term “progressively”. The sentence “Notably, ROS production progressively increases with age, consistent with the severity of the disease phenotype.” has been modified to “**Notably, ROS production increased with age, consistent with the severity of the disease phenotype**” (page 14).

21) *How do you separate heart rhythm verses contraction from a florescence image movie? This is not clear it can be done.*

Response: We thank the referee for reviewing our study’s methodology for separating cardiac rhythm from contraction. To assess cardiac function, we referred to the original paper for this method (Sampurna et al., *Inventions* 2018). The separation of heart rhythm from contraction is achieved through a multi-step process.

(1) Video capture: In vivo videos of the beating hearts were recorded using a fluorescence stereomicroscope (Biorevo BZ-9000, KEYENCE).

(2) Image analysis: The captured videos were imported into the ImageJ software, and the EGFP fluorescence signal was used to outline the heart region.

(3) Region of Interest (ROI) selection: The circle tool in ImageJ was used to select the same ROI in the atrial region.

(4) Dynamic pixel changes: The resulting plot profile was analyzed to obtain dynamic pixel changes. This analysis allowed us to distinguish and quantify the cardiac rhythm and contraction of the zebrafish in each image of the video stack.

By following these steps, we were able to separately assess and quantify heart rhythm and contraction, thereby providing a comprehensive understanding of cardiac function in the zebrafish. We have added this information to the Methods section (pages 26–27).

22) *Regarding stats – are all data normally distributed, any outliers? ANOVA may not be best if not a normal distribution. Is a power test done to detect if all sample sizes are big enough?*

Response: We acknowledge the oversight in choosing statistical tests. Based on our

reevaluation, non-parametric tests are an appropriate choice for several datasets because they do not follow a normal distribution [Fig. 4n (Il-1 β , TNF α); Fig. 5a, h, j; Supplementary Figure 7c, d; Supplementary Figure 9 (srf3a, rpl2211, rpl10a, rassf1, atf4)]. To address this issue, we thoroughly reassessed our statistical methods and incorporated non-parametric tests into our analysis (page 33).

Regarding the sample size in our study, we acknowledge that a power analysis was not initially performed. This decision was influenced by the zebrafish model that, unlike mice, does not have inbred lines, which poses a challenge to genetic consistency. To mitigate this, we used mutant zebrafish and their wild-type siblings from the same mother fish to minimize potential genetic background differences that could affect the phenotype. In addition, when larvae are required, the stress caused by genotyping can be lethal, limiting the available sample size. We recognize that no established guidelines or standards currently exist for conducting power analyses specifically for zebrafish research. This methodology warrants further investigation and development. We understand the limitations of our current methodology and acknowledge the need for more detailed validation of our experimental results in future studies. We believe that this approach balances the ethical considerations of using an animal with the need for scientific rigor.

23) Fig 3 – regarding positive area of the brain, how was this measured, just across the dorsal surface? Are there better images to show this? Tissue sections would be better at equal positions or across an equal number/size of brain sections.

Response: We have included frozen sections and SA- β -galactosidase staining to provide a more accurate representation of the positive brain areas. Quantification was performed by measuring the positive areas across equal brain sections to ensure the consistency and reliability of our analysis (Fig. 4g-l).

24) Were male verses female differences by 3 mos and beyond done to account for any sex differences in smg9 function and NMD?

Response: While the current study focused primarily on male zebrafish, we

acknowledge the importance of investigating potential sex-specific differences in smg9 function and NMD. In future studies, we plan to examine both male and female zebrafish to elucidate sex-related differences. We have added “All adult fish used in this study were male.” to the Methods section (page 25).

Point-by-point replies to Referee #4

Summary:

This manuscript sought to define the role of nonsense-mediated decay (NMD) factor SMG9 in zebrafish. The authors used a CRISPR/Cas9 approach to generate a line of mutant fish, which displayed defects in brain size, cardiac contractility, and induced premature aging phenotypes. They attributed these phenotypes to overexpression of the smox RNA, which the authors show is upregulated in SMG9 mutant fish. The results of the study are interesting and novel, however this reviewer finds a conceptual flaw in the design and description of the smg9^{-/-} as a “null”, as described below. Please address this and the other concerns outlined below.

Major concern:

1. The authors generated a “frameshift deletion” in exon 1, which they state, “likely corresponded to a null allele”. However, the authors fail to experimentally demonstrate this is a null mutation. This is essential to describe the mutants as “null”.

A search of Ensembl by this reviewer found that Danio rerio has two isoforms for smg9, which differ by the first two exons (absent in coding for one isoform). A frameshift in exon 1, as the authors have done, likely only disrupts one of the two isoforms and may be a hypomorphic allele rather than a null, which may account for the difference in embryonic lethality compared to the mouse knockouts that the authors reference. The authors must address this issue experimentally (for example, using qRT-PCR or Western blot).

Response: We appreciate the careful evaluation of our manuscript and thank the referees for providing constructive feedback. We have considered the concern regarding characterizing the smg9^{-/-} allele as a “null” mutation.

The anti-human SMG9 polyclonal antibody from Bethyl Laboratories reacted with the zebrafish Smg9. Therefore, we performed a western blot analysis and confirmed the absence of Smg9 protein expression (Fig. 1b). A minor isoform reported in Ensembl was not observed, indicating that this protein isoform was not expressed in zebrafish (**Supplementary Fig. 2a**). We have registered this smg9 mutant zebrafish with ZFIN as

smg9^{oi7/oi7} and refer to it as such in the revised manuscript.

Additional concerns:

1. *Authors do not adequately address/discuss differences in phenotypes observed between this study and those in mouse (knockout lethal vs viable) and human (function of SMG8, SMG9 as a positive vs negative regulator of UPF1 phosphorylation).*

Response: We thank the referee for their valuable comment.

Regarding differences in phenotypes observed between *smg9^{oi7/oi7}* and *Smg9*-knockout mice (knockout lethal vs. viable), we have confirmed that both isoforms are deleted in *smg9^{oi7/oi7}* zebrafish. Therefore, the phenotypic differences between mice and zebrafish models are not caused by the residual activity of another isoform. Therefore, assuming that embryonic lethality in *Smg9* knockout mice is caused by placental defects is reasonable. We have added this to the Discussion section (pages 18).

Regarding the differences in phenotypes observed between *smg9^{oi7/oi7}* and humans, SMG8 and SMG9 function as positive vs. negative regulators of UPF1 phosphorylation. Except for the testes, SMG9 mRNA expression showed approximately the same pattern in humans and zebrafish, suggesting the phenotypic differences between zebrafish and humans are not caused by differences in gene expression. Because the mutation in human patients results in a missense mutation rather than a null mutation, as in our *smg9^{oi7/oi7}* model, a residual hypomorphic function may still be present. We have added this information on page 4.

Regarding the function of SMG8 and SMG9 as positive and negative regulators of UPF1 phosphorylation, the SMG8:SMG9 heterodimer shows different effects on UPF1 phosphorylation in vitro and in vivo. We examined whether human and zebrafish SMG8 and SMG9 are interchangeable in the SMG1:SMG8:SMG9 complex. First, we overexpressed the human SMG8:SMG9 heterodimers in 293T cells (Supplementary Figure 10b). As reported (Yamashita et al., *Genes Dev* 2009), the human SMG8:SMG9 heterodimer suppressed SMG1 activity in vitro. This effect was also observed upon overexpression of the zebrafish *Smg8:Smg9* heterodimer. Furthermore, human SMG8

and zebrafish SMG9 heterodimers, and conversely, zebrafish SMG8 and human SMG9 heterodimers, inhibited the activity of human SMG1 in UPF1 phosphorylation in vitro. These results indicate that the components of the SMG1:SMG8:SMG9 complex are interchangeable between humans and zebrafish. However, this result conflicts with the decrease in UPF1 phosphorylation in *smg9^{Goi7/oi7}* zebrafish in vivo (Fig. 5k, l) (pages 12–13).

The different effects of SMG8/9 on SMG1 activity in vitro and in vivo may be caused by differences in spatial and temporal regulation. SMG9 is involved in the stability of SMG1 and the efficiency with which the SMG8/9 and SMG1 protein complex is recruited to the EJC (Arias-Palomo et al., *Genes Dev* 2011). Indeed, reports have shown that NMD itself is positively regulated by SMG8/9 in vivo, even though SMG8/9 suppresses SMG1 phosphorylation in vitro (Yamashita et al., *Genes Dev* 2009). Further mechanistic analyses are required to clarify the relationships between SMG8, SMG9, and SMG1 (page 22).

2. In Introduction, bottom of pg. 2, statement “SMG9 presumably mediates the interaction between SMG8 and SMG1.” needs reference(s).

Response: We thank the referee for bringing this point to our attention. We have added the following references:

1. Arias-Palomo, E. et al. The nonsense-mediated mRNA decay SMG-1 kinase is regulated by large-scale conformational changes controlled by SMG-8. *Genes & Dev.* 25, 153-164 (2011)

2. Fernandez, I. S. et al. Characterization of SMG-9, an essential component of the nonsense-mediated mRNA decay SMG1C complex. *Nucleic Acids Res.* 39, 347-358 (2011).

3. To call *smg8* mutant a “null”, needs experimental validation it is a true null (qRT-PCR or Western blot). Otherwise, should just call “mutant” in this case.

Response: As the referee suggested, the null mutation in *smg8^{-/-}* zebrafish must be

validated using western blotting. However, we were unable to obtain an antibody that could react with the zebrafish Smg8. Therefore, RT-qPCR was performed to confirm the absence of wild-type *smg8* mRNA (Supplementary Fig. 3c). Because the possibility of hypomorphism cannot be dismissed completely, labeling it as “mutant” rather than “null” is appropriate, as the referee advised (page 6). The *smg8* mutant zebrafish was registered in ZFIN as *smg8*^{oi8/oi8} and is referred to as such in the revised manuscript.

4. Including representative time lapse movies of cardiac contraction as depicted in Fig. 2 in still and quantification in Supplementary material would be helpful to illustrate differences in contractility.

Response: We appreciate the referee’s suggestion to include representative time-lapse movies of cardiac contraction. We have included such movies (Supplementary Video). These videos provide a dynamic representation of cardiac contraction, complementing the static images presented in Figure 2. We believe this addition enhances the clarity and completeness of our findings.

5. For statistical analyses, authors must confirm data are normally distributed for parametric tests to be appropriate. Otherwise, nonparametric tests should be used.

Response: We thank you for your feedback regarding the statistical methods used in this study. We acknowledge the oversight regarding the choice of statistical tests. Based on our reevaluation, non-parametric tests are appropriate for several datasets because they do not follow a normal distribution [Fig. 4n (Il-1 β , TNF α); Fig. 5a, h, j; Supplementary Figure 7c, d; Supplementary Figure 9 (*srf3a*, *rpl2211*, *rpl10a*, *rassf1*, *atf4*)]. To address this issue, we thoroughly reassessed our statistical methods and incorporated non-parametric tests into our analysis (page 33).

6. There is odd spacing in Fig. 3a—(*n*=2 0) instead of (*n*=20).

Response: We have corrected the errors in Figure 3a in the revised manuscript.

7. *In Fig. 3d,f, images look excessively manipulated to remove background around the fish. It is unclear why images weren't taken similarly to Fig. 3b,c. Also, it is difficult to tell sex of fish in images because the dorsal and anal fins are folded, or possibly due to how the images are manipulated?*

Response: A high-intensity light source was used to visualize the SA- β -gal staining in the image. This choice was an unintended consequence of overexposing the background of the image. The images were not manipulated using digital editing tools. We have selected a more natural and easily sex-distinguishable alternative to the previous images (Fig. 3b, c).

8. *Scale bars missing in Fig. 3b,c.*

Response: We have made the appropriate corrections.

9. *Description of SA- β -gal signal quantification in Methods is inadequate—please provide better detail regarding how thresholding was performed.*

Response: We have revised the "Senescence-associated β -galactosidase (SA- β -gal) staining" section in the Methods to include additional details on how the thresholding was performed (pages 27–28).

SA- β -gal signals were quantified using Image J's color threshold selection tool, version 1.52a (National Institutes of Health; Bethesda, MD, USA). A precise thresholding process was employed to define the specific color intensity ranges corresponding to positive SA- β -gal staining while excluding background noise. The lower and upper bounds of the threshold were carefully selected to guarantee precise and repeatable quantification of each sample. SA- β -gal activity was measured by calculating the percentage of positively stained areas relative to the entire area of interest.

10. *Images in Supp Fig 6 need to be better labeled for marker/color, and channels should be separated—it's difficult to discern differences in HuC/D as displayed.*

Response: We have prepared individual images for each channel to facilitate better discernment of differences in HuC/D staining (Supplementary Figure 7a)

11. Full blot image in Supplement Fig. 9 should have wells labeled.

Response: We thank the referee for their suggestion. We have added labels to the full blot image for better clarity and reference (Supplementary Figure 20).

12. The authors do not adequately describe/indicate the sex of the adult fish used in the study.

Response: All adult fish used in this study were male. We have added this information in the Methods section (page 25).

REVIEWERS' COMMENTS:

Reviewer #1 (Remarks to the Author):

I would like to thank the authors for their additional analyses which bring more evidence and strengthen their initial claim. Congratulations on a great manuscript.

Reviewer #3 (Remarks to the Author):

Most of my comments were fully addressed, however, there are four things in the initial set of comments that I still have questions/concerns about as outlined below. I included my original comment, the author response, and then below each my additional comment following the response.

5) The early smox mRNA increase experiment cannot be linked to later aging without making further observations plus use of other molecular tools. Observing acute defects on heart and brain and rescuing these with smox inhibition alone are not enough; observe aging or aging markers later.
Response: To address this concern, we performed an additional experiment in which zebrafish smox mRNA was injected into single-cell-stage embryos. We observed and assessed aging markers, including lifespan and fertility, in adult fish to gain a better understanding of the long-term effects of Smox modulation (Fig. 6f, Supplementary Figure 14) (page 14). Because overexpression of Smox during only the larval stage increases SA- β -gal staining in the brains of adult fish, accumulation of ROS during the larval stage should be sufficient to induce an early aging phenotype (Fig. 6f) (page 14). We hope this additional analysis clarifies the link between early molecular changes and later-aging outcomes in our study. However, the lifespan and fertility of smox mRNA injected zebrafish were similar to those of the control group, indicating that short-term smox expression is insufficient to induce full aging phenotypes. Investigating whether Smox upregulation is required for the premature aging phenotypes in smg9oi7/oi7 zebrafish has an experimental limitation because long-term treatment with SMOX and ROS inhibitors is impossible due to the toxicity of these inhibitors. Generating smg9oi7/oi7 zebrafish with a smox deficiency would be useful for testing this hypothesis.
My added comment: This added experiment seems mostly sufficient to address my concerns; however, I am not certain that the authors fully understand that the limitations of single one-cell stage zebrafish mRNA microinjection. They are to a 2-3 day time period of activity of the mRNA and translated protein at most unless they have validation of longer activity; it is unlikely that the smox mRNA will still be available and translated "...during only the larval stage" as they wrote. My original comment was related to this experimental limitation, and I am not sure that the authors are also fully aware of this or not or if they can better address this in the text. Larval stage is after 72 hpf, when this mRNA would no longer be functional.

7) How much smg9 mRNA was used for the rescue? Methods only state that – "Each mRNA (400 ng/ μ L) was injected into one-cell-stage embryos" How much per embryo or ng per embryo?

Response: We apologize for the inadequate description in the Methods section. We have added the following statement (page 26): Forty picograms of each mRNA were injected into individual embryos at the single cell stage.

My added comment: Thank you for indicating the total mRNA amount per embryo, but for me the math does not align between what was originally written and this correction – was 400ng/ μ L the stock mRNA concentration or the final concentration of mRNA in the injection mix. Typically, 1-4 nL total is injected per embryo, so this needs to be more fully explained still so that the math lines up between injection mix mRNA concentration (ng/ μ L), injection amount per embryo (nL), and total mRNA per embryo (pg).

22) Regarding stats – are all data normally distributed, any outliers? ANOVA may not be best if not a normal distribution. Is a power test done to detect if all sample sizes are big enough? Response: We acknowledge the oversight in choosing statistical tests. Based on our reevaluation, non-parametric

tests are an appropriate choice for several datasets because they do not follow a normal distribution [Fig. 4n (Il-1 β , TNF α); Fig. 5a, h, j; Supplementary Figure 7c, d; Supplementary Figure 9 (srf3a, rpl22l1, rpl10a, rassf1, atf4)]. To address this issue, we thoroughly reassessed our statistical methods and incorporated non-parametric tests into our analysis (page 33). Regarding the sample size in our study, we acknowledge that a power analysis was not initially performed. This decision was influenced by the zebrafish model that, unlike mice, does not have inbred lines, which poses a challenge to genetic consistency. To mitigate this, we used mutant zebrafish and their wild-type siblings from the same mother fish to minimize potential genetic background differences that could affect the phenotype. In addition, when larvae are required, the stress caused by genotyping can be lethal, limiting the available sample size. We recognize that no established guidelines or standards currently exist for conducting power analyses specifically for zebrafish research. This methodology warrants further investigation and development. We understand the limitations of our current methodology and acknowledge the need for more detailed validation of our experimental results in future studies. We believe that this approach balances the ethical considerations of using an animal with the need for scientific rigor.

My added comment: To my knowledge there is not a requirement or standard in any field for a power analysis; this is a statistical test used to determine if the sample size analyzed was large enough to determine whether or not a significant difference could have been observed between groups. I still believe it is possible in this case, and it should be done to determine if sample sizes for these experiments were reliable enough to make the conclusions presented in this paper, or the authors could acknowledge somehow that they chose to leave this out and keep sample sizes perhaps less than ideal.

23) Fig 3 – regarding positive area of the brain, how was this measured, just across the dorsal surface? Are there better images to show this? Tissue sections would be better at equal positions or across an equal number/size of brain sections.

Response: We have included frozen sections and SA- β -galactosidase staining to provide a more accurate representation of the positive brain areas. Quantification was performed by measuring the positive areas across equal brain sections to ensure the consistency and reliability of our analysis (Fig. 4g-l).

My added comment: From the images shown, I am still not certain how a positive brain area was measured – what software was used for this? Was it quantity of blue pixels? Most of these images look light to dark blue. Could the authors use brackets or arrows to indicate what is positive versus negative expression? The images are blurry in my view.

Reviewer #4 (Remarks to the Author):

This manuscript sought to define the role of nonsense-mediated decay (NMD) factor SMG9 in zebrafish. The authors used a CRISPR/Cas9 approach to generate a line of mutant fish, which displayed defects in brain size, cardiac contractility, and induced premature aging phenotypes. They attributed these phenotypes to overexpression of the smox RNA, which the authors show is upregulated in SMG9 mutant fish. The results of the study are interesting and novel.

The authors carefully and thoroughly addressed all of my prior concerns. I have no outstanding concerns.

Reply to Referee #1

I would like to thank the authors for their additional analyses which bring more evidence and strengthen their initial claim. Congratulations on a great manuscript.

Response: We deeply appreciate your kind words and acceptance of our manuscript. We thank you and your constructive feedback, which has greatly helped to improve the quality and impact of our work. Thank you very much for your time and expertise.

Point-by-point replies to Referee #3

Thank you for your additional comments. We greatly appreciate the detailed feedback and are addressing the four concerns you raised in our revised manuscript. Below, we provide point-by-point responses to each comment and have made the corresponding revisions to the manuscript, which have been highlighted in yellow for easy identification.

Most of my comments were fully addressed, however, there are four things in the initial set of comments that I still have questions/concerns about as outlined below. I included my original comment, the author response, and then below each my additional comment following the response.

5) The early smox mRNA increase experiment cannot be linked to later aging without making further observations plus use of other molecular tools. Observing acute defects on heart and brain and rescuing these with smox inhibition alone are not enough; observe aging or aging markers later.

Response: To address this concern, we performed an additional experiment in which zebrafish smox mRNA was injected into single-cell-stage embryos. We observed and assessed aging markers, including lifespan and fertility, in adult fish to gain a better understanding of the long-term effects of Smox modulation (Fig. 6f, Supplementary Figure 14) (page 14). Because overexpression of Smox during only the larval stage increases SA- β -gal staining in the brains of adult fish, accumulation of ROS during the larval stage should be sufficient to induce an early aging phenotype (Fig. 6f) (page 14). We hope this additional analysis clarifies the link between early molecular changes and later-aging outcomes in our study. However, the lifespan and fertility of smox mRNA injected zebrafish were similar to those of the control group, indicating that short-term smox expression is insufficient to induce full aging phenotypes. Investigating whether Smox upregulation is required for the premature aging phenotypes in smg9oi7/oi7 zebrafish has an experimental limitation because long-term treatment with SMOX and ROS inhibitors is impossible due to the toxicity of these inhibitors. Generating smg9oi7/oi7 zebrafish with a smox deficiency would be useful for testing this hypothesis.

My added comment: This added experiment seems mostly sufficient to address my

concerns; however, I am not certain that the authors fully understand that the limitations of single one-cell stage zebrafish mRNA microinjection. They are to a 2-3 day time period of activity of the mRNA and translated protein at most unless they have validation of longer activity; it is unlikely that the smox mRNA will still be available and translated "...during only the larval stage" as they wrote. My original comment was related to this experimental limitation, and I am not sure that the authors are also fully aware of this or not or if they can better address this in the text. Larval stage is after 72 hpf, when this mRNA would no longer be functional.

Response: Thank you for your insightful feedback regarding the duration of *smox* mRNA activity following a single-cell stage zebrafish mRNA microinjection. We acknowledge and regret the error in our reply where we stated that *smox* mRNA would be active "during only the larval stage." It should be noted that the activity of the injected mRNA typically lasts for only 2-3 days post-injection, and does not extend into the larval stage that begins after 72 h. We recognize this as a significant experimental limitation and appreciate your clarity in highlighting this issue. In future studies, we plan to address this by generating *smg90* zebrafish with a *smox* deficiency, which will allow us to more accurately delineate the role of *smox* in the aging process.

7) How much smg9 mRNA was used for the rescue? Methods only state that – "Each mRNA (400 ng/μL) was injected into one-cell-stage embryos" How much per embryo or ng per embryo?

Response: We apologize for the inadequate description in the Methods section. We have added the following statement (page 26): Forty picograms of each mRNA were injected into individual embryos at the single cell stage.

My added comment: Thank you for indicating the total mRNA amount per embryo, but for me the math does not align between what was originally written and this correction – was 400ng/μL the stock mRNA concentration or the final concentration of mRNA in the injection mix. Typically, 1-4 nL total is injected per embryo, so this needs to be more fully explained still so that the math lines up between injection mix mRNA concentration (ng/μL), injection amount per embryo (nL), and total mRNA per embryo (pg).

Response: Thank you for your attention to the details regarding the mRNA concentration in the injection mix and total mRNA amount per embryo. We apologize for the confusion caused by the initial error in stating the mRNA concentration. The correct mRNA concentration in the injection mix was 40 ng/ μ L, not 400 ng/ μ L, as previously stated.

Each embryo was injected with 1 nL of the mRNA solution. Therefore, the total mRNA per embryo is calculated as follows:

$$\begin{aligned} \text{Total mRNA per embryo} &= \text{mRNA concentration} \times \text{injection volume per embryo} \\ &= 40 \text{ ng}/\mu\text{L} \times 1 \text{ nL} = 40 \text{ pg} \end{aligned}$$

Thus, the corrected total mRNA level per embryo is 40 pg.

We appreciate your guidance for ensuring the accuracy of our method. We have revised the manuscript to reflect this clarification (Page 26).

22) Regarding stats – are all data normally distributed, any outliers? ANOVA may not be best if not a normal distribution. Is a power test done to detect if all sample sizes are big enough? Response: We acknowledge the oversight in choosing statistical tests. Based on our reevaluation, non-parametric tests are an appropriate choice for several datasets because they do not follow a normal distribution [Fig. 4n (Il-1 β , TNF α); Fig. 5a, h, j; Supplementary Figure 7c, d; Supplementary Figure 9 (srf3a, rpl221l, rpl10a, rassf1, atf4)]. To address this issue, we thoroughly reassessed our statistical methods and incorporated non-parametric tests into our analysis (page 33). Regarding the sample size in our study, we acknowledge that a power analysis was not initially performed. This decision was influenced by the zebrafish model that, unlike mice, does not have inbred lines, which poses a challenge to genetic consistency. To mitigate this, we used mutant zebrafish and their wild-type siblings from the same mother fish to minimize potential genetic background differences that could affect the phenotype. In addition, when larvae are required, the stress caused by genotyping can be lethal, limiting the available sample size. We recognize that no established guidelines or standards currently exist for conducting power analyses specifically for zebrafish research. This methodology warrants further investigation and development. We understand the limitations of our current methodology and acknowledge the need for more detailed validation of our experimental results in future studies. We believe that this approach balances the ethical considerations of using an animal with the need for scientific rigor.

My added comment: To my knowledge there is a not a requirement or standard in any

field for a power analysis; this is a statistical test used to determine if the sample size analyzed was large enough to determine whether or not a significant difference could have been observed between groups. I still believe it is possible in this case, and it should be done to determine if sample sizes for these experiments were reliable enough to make the conclusions presented in this paper; or the authors could acknowledge somehow that they chose to leave this out and keep sample sizes perhaps less than ideal.

Response: We appreciate your feedback regarding the use of power analysis to validate our sample sizes. In response, we have carefully performed a power analysis using the latest version of G*Power software 3.1

(<https://www.psychologie.hhu.de/arbeitsgruppen/allgemeine-psychologie-und-arbeitspsychologie/gpower>). The statistical power of the study was set at 0.8. This analysis confirmed that the sample sizes for the majority of our experiments were sufficient to reliably detect significant differences between groups. However, we found that the sample sizes for the *srsf3a* and *smox* RT-qPCR experiments, as shown in Supplementary Figure 9, were indeed insufficient.

To correct this, we have increased the sample size from 6 to 8 in both cases. We have added a note in the Statistics and Reproducibility section of the Methods that the power analysis was performed using G*Power software (Page 34). We recognize the critical importance of adequate sample sizes in drawing robust conclusions and thank you for highlighting this point.

23) Fig 3 – regarding positive area of the brain, how was this measured, just across the dorsal surface? Are there better images to show this? Tissue sections would be better at equal positions or across an equal number/size of brain sections.

Response: We have included frozen sections and SA-β-galactosidase staining to provide

a more accurate representation of the positive brain areas. Quantification was performed by measuring the positive areas across equal brain sections to ensure the consistency and reliability of our analysis (Fig. 4g-l).

My added comment: From the images shown, I am still not certain how a positive brain area was measured – what software was used for this? Was it quantity of blue pixels? Most of these images look light to dark blue. Could the authors use brackets or arrows to indicate what is positive verses negative expression? The images are blurry in my view.

Response: Thank you for your comment. We appreciate your attention to the details regarding the visualization of positive regions of brain sections. We used Image J software for this analysis, specifically applying the "Image > Adjust > Color Threshold" function to standardize saturation and brightness across all images. This adjustment transformed the stained areas to a red color, which facilitated the differentiation between positive and negative regions. The enhanced distinction is illustrated in the figure below, which are from the Figure 4k. Arrows have been added to Figures 4i and 4k to indicate the positive regions (Page 43).

Reply to Referee #4

This manuscript sought to define the role of nonsense-mediated decay (NMD) factor SMG9 in zebrafish. The authors used a CRISPR/Cas9 approach to generate a line of mutant fish, which displayed defects in brain size, cardiac contractility, and induced premature aging phenotypes. They attributed these phenotypes to overexpression of the smox RNA, which the authors show is upregulated in SMG9 mutant fish. The results of the study are interesting and novel.

The authors carefully and thoroughly addressed all of my prior concerns. I have no outstanding concerns.

Response: We sincerely appreciate your thoughtful and detailed review of our manuscript. Your comments were invaluable in refining our study and clarifying our findings. Thank you again for your time and expertise.